# DeepWeightFlow: Re-Basined Flow Matching for Generating Neural Network Weights

**Saumya Gupta, Scott Biggs, Maritz Laber, Zohair Shafi, and Ayan Paul**
Northeastern University
Boston MA 02119, USA
`{gupta.saumy,biggs.s,laber.m,shafi.z,r.walters,a.paul}@northeastern.edu`

## Abstract

Building efficient and effective generative models for neural network weights has been a research focus of significant interest that faces challenges posed by the high-dimensional weight spaces of modern neural networks and their symmetries. Several prior generative models are limited to generating partial neural network weights, particularly for larger models, such as ResNet and ViT. Those that do generate complete weights struggle with generation speed or require finetuning of the generated models. In this work, we present DeepWeightFlow, a Flow Matching model that operates directly in weight space to generate diverse and high-accuracy neural network weights for a variety of architectures, neural network sizes, and data modalities. The neural networks generated by DeepWeightFlow do not require fine-tuning to perform well and can scale to large networks. We apply Git Re-Basin and TransFusion for neural network canonicalization in the context of generative weight models to account for the impact of neural network permutation symmetries and to improve generation efficiency for larger model sizes. The generated networks excel at transfer learning, and ensembles of hundreds of neural networks can be generated in minutes, far exceeding the efficiency of diffusion-based methods. DeepWeightFlow models pave the way for more efficient and scalable generation of diverse sets of neural networks.

## 1 Introduction

Generating neural network weights is a sampling challenge that explores the underlying high-dimensional distribution of weights, where neural networks trained on similar datasets and tasks exhibit statistical regularities. The development of generative models capable of learning the distributional properties of trained weights faces challenges of symmetries and high-dimensionality of the weight spaces. Treating large collections of neural network weights as a structured and high-dimensional data modality promises advances in model editing (Mitchell et al., 2022; Meng et al., 2022), accelerating transfer learning (Knyazev et al., 2021; Schürholt et al., 2022), facilitating uncertainty quantification (Lakshminarayanan et al., 2017), and advancing neural architecture search (Chen et al., 2019; Chen, 2023). Unlike traditional machine learning tasks that aim to optimize weights for specific downstream tasks, this concept advocates sampling from the weight space itself. *In this work, we focus on the efficient generation of complete neural network weights that can achieve high performance for a given task and excel at transfer learning* thus addressing fundamental limitations in current deep learning workflows, such as computational bottlenecks in iterative training, vulnerability to adversarial attacks (Goodfellow et al., 2015; Madry et al., 2018) and privacy concerns arising from training data reconstructions (Nasr et al., 2019; Tramer et al., 2022).

Generating neural network weights faces three main challenges: *Firstly*, neural network weights have a rich class of symmetries (Hecht-Nielsen, 1990; Entezari et al., 2022; Navon et al., 2023; Zhao et al., 2025), i.e., transformations of the weights that leave the neural network functionally invariant. Most prominently, joint permutations of hidden neurons in adjacent layers of multi-layer perceptrons (MLP) do not change the encoded function. Other architectural choices, such as incorporating attention heads or the choice of non-linear activation, can induce additional symmetries. Techniques for dealing with weight space symmetries fall into three main categories: (1) data augmentation, (2) equivariant architectures, and (3) canonicalization. Prior work, such as Wortsman et al. (2022);

**DeepWeightFlow Pipeline**

**a)** Training data     **b)** Canonicalization     **c)** Weight generation

Target networks     (optional)

$f(x \mid W_1, \dots, W_L)$     $\tilde{W}_i = P_i W_i P^{\mathsf{T}}$     $(W'_1, \dots, W'_L) \sim p_{\hat{\theta}}$

Flow Matching $p_{\hat{\theta}}$

$W_1 \quad W_2 \quad \cdots \quad W_L$     $\tilde{W}_1 \quad \tilde{W}_2 \quad \cdots \quad \tilde{W}_L$     $W'_1 \quad W'_2 \quad \cdots \quad W'_L$

Figure 1: *Schematic depiction of DeepWeightFlow.* **a)** *We construct a training dataset of weights by fully training neural networks with weights $W_1, \dots, W_L$ on a given target task.* **b)** *Optionally, we use canonicalization, i.e., choosing a canonical representative $\tilde{W}_i$ from the same orbit as $W_i$, to break the permutation symmetry in parameter space.* **c)** *We train a flow model $p_{\hat{\theta}}$ for efficient generation of high-performance weights $(W_1, \dots, W_L) \sim p_{\hat{\theta}}$ for the target task.*

Wang et al. (2024); Soro et al. (2025); Saragih et al. (2025a), does not actively account for symmetries in their generative models, while others, such as Saragih et al. (2025b), use equivariant architectures. Data augmentation has also been explored in weight representation learning (Schürholt et al., 2024; Shamsian et al., 2023; 2024), and to a lesser extent in weight generation (Schürholt et al., 2024; Wang et al., 2025). Finally, canonicalization has recently found application in weight space learning (Schürholt et al., 2024; Wang et al., 2024; 2025), borrowing ideas from model merging and alignment (Ainsworth et al., 2023; Rinaldi et al., 2025). *Secondly*, neural network weights are high-dimensional, varying from tens of millions for a small ResNet (He et al., 2016) to hundreds of billions for modern large language models (Touvron et al., 2023; Guo et al., 2025). This challenge is often addressed by non-linear, dimensionality reduction techniques, including variational autoencoders (VAEs) (Soro et al., 2025) and graph autoencoders (Schürholt et al., 2022; Saragih et al., 2025b; Soro et al., 2025). Despite increasing efficiency, dimensionality reduction requires training an additional model for dimensionality reduction and can be detrimental to the quality of the generated weights if the compression is lossy. *Lastly*, generative models proposed recently either generate partial weights for large models, or require finetuning post-generation, or have long generation time per sample, making them impractical.

To address these challenges, we propose DeepWeightFlow, a method for efficient generation of high-performance neural network weights via Flow Matching (FM) and apply it to MLP for vision and tabular data, as well as ResNet (He et al., 2016), and ViT (Dosovitskiy et al., 2021) for computer vision tasks, and BERT for natural language processing (NLP) (Devlin et al., 2019). We rely on canonicalization techniques, such as Git Re-Basin (Ainsworth et al., 2023) and TransFusion (Rinaldi et al., 2025), to resolve parameter permutation symmetries, and show that canonicalization aids weight generation for large neural networks but offers limited benefits when the weight space dimension is moderate. We show that neural networks generated by DeepWeightFlow excel at the target task and are competitive with state-of-the-art weight generation methods such as RPG (Wang et al., 2025), D2NWG (Soro et al., 2025), FLoWN (Saragih et al., 2025b), and P-diff (Wang et al., 2024) while overcoming several of the limitations of these models. A schematic of our methods is shown in Figure 1. While DeepWeightFlow samples directly from weight spaces, we show that the models can scale to generating larger networks using PCA while keeping the training and the generation time low. *In summary, the contributions of this work are as follows:*

- DeepWeightFlow is a new method for *complete* neural network weight generation based on FM, unconditioned by dataset characteristics, task descriptions, or architectural specifications. DeepWeightFlow does not require additional training of autoencoders for dimensionality reduction and can scale to high-dimensional weight spaces using PCA.
- We show that our method can generate weights for neural networks with $\mathcal{O}(100M)$ parameters, and diverse architectures, such as MLP, ResNet, ViT, and BERT that, without fine-tuning, exhibit high performance on tasks in the vision, tabular, and natural language domains.
- We empirically elucidate the role of parameter symmetry for weight generation, showing that canonicalization of the training data aids the generation of very high-dimensional weights but offers no additional benefit for weights of modest dimension.
- DeepWeightFlow, with a simple MLP implementation, and without any equivariant architecture, is far more efficient in generating diverse samples compared to diffusion-based models.

## 2 RELATED WORK

**HyperNetworks:** Early explorations of neural network generation focus on HyperNetworks, which learn neural network parameters as a relaxed temporal weight sharing process (Ha et al., 2017). HyperNetworks have been applied to generating weights through density sampling, GAN, and diffusion methods by learning latent representations of neural network weights (Ha et al., 2017; Frankle & Carbin, 2019; Ratzlaff & Fuxin, 2019; Schürholt et al., 2022; Kiani et al., 2024). They have also been used to build meta-learners – augmentations or substitutes for Stochastic Gradient Descent optimization, which condition generation of new weight checkpoints on prior weights and task losses (Peebles et al., 2022; Zhang et al., 2024; Wang et al., 2025).

**Generative Models for Neural Network Weights:** Diffusion-based generative models for weights have been successful at neural network weight generation, but often do not directly resolve weight space symmetries. These approaches either provide no treatment (Wang et al., 2024), or rely on Variational Auto Encoding (VAE) methods to concurrently resolve weight symmetries and reduce the dimensionality of the generative task (Ha et al., 2017; Frankle & Carbin, 2019; Schürholt et al., 2022; Kiani et al., 2024; Soro et al., 2025). In contrast, weight canonicalization is done as a pre-training step in SANE (Schürholt et al., 2024), which uses kernel density sampling of hypernetwork latents to autoregressively populate models layer-wise, allowing for *complete* weight generation, but requires fine-tuning, unlike DeepWeightFlow. Diffusion has been applied directly to generating partial (Wang et al., 2024) or complete weights (Soro et al., 2025; Wang et al., 2025). RPG (Wang et al., 2025) generates complete weights by using a recurrent diffusion model. However, RPG shows long generation times, often taking hours to generate a set of networks that DeepWeightFlow takes minutes to complete. Subsequent Conditional Flow Matching (CFM) methods (Saragih et al., 2025b;a) explore dataset embeddings as conditioning for transfer learning and weight generation. These CFMs also report using VAE methods to reduce the dimensionality of the generative task and to resolve weight symmetries (Saragih et al., 2025b;a). We develop this further with DeepWeightFlow, which operates directly in deep weight space to generate *complete* weight sets, and demonstrates the viability of PCA as a strategy for surpassing $\mathcal{O}(100M)$ parameter sets.

**Permutation Symmetries in Weight Space:** SANE (Schürholt et al., 2024) applies Git Re-Basin as a canonicalization for hypernetwork training (Schürholt et al., 2022; 2024; Ainsworth et al., 2023). Unlike DeepWeightFlow, SANE tokenizes weights layer-wise and autoregressively samples them to populate new neural models. RPG (Wang et al., 2025) uses a different strategy to address permutation symmetry by one-hot encoding models to differentiate between potential permutations of similar weights. D2NWG (Soro et al., 2025) and FLoWN (Saragih et al., 2025b) both evaluate VAEs, while FLoWN additionally considers permutation invariant graph autoencoding methods to appeal to the manifold and lottery ticket hypotheses (Ha et al., 2017; Frankle & Carbin, 2019; Schürholt et al., 2022; Kiani et al., 2024). DeepWeightFlow extends the canonicalization methods from previous works to transformers through TransFusion, and thoroughly evaluates the impact of canonicalization on generating *complete* weight sets (Schürholt et al., 2024; Wang et al., 2024; 2025; Soro et al., 2025).

## 3 BACKGROUND

DeepWeightFlow is an FM model using an MLP architecture trained on canonicalized neural networks. In this section, we give a brief overview of the various methods we use to build it.

### 3.1 FLOW MATCHING

Flow Matching (Lipman et al., 2023) is a generative technique for learning a vector field to transport a noise vector to a target distribution. Given an unknown data distribution $q(x)$, we define a probability path $p_t$ for $t \in [0, 1]$ with $p_0 \sim \mathcal{N}(0, 1)$ and $p_1 \approx q(x)$. FM learns a vector field with parameters $\theta$, $v_\theta(x, t)$, that transports $p_0$ to $p_1$ by minimizing

$$\mathcal{L}_{\text{FM}}(\theta) = \mathbb{E}_{t \sim \mathcal{U}[0,1], x \sim p_t(x)}\big[\|v_\theta(x, t) - u(x, t)\|^2\big], \tag{1}$$

where $u(x, t)$ is the true vector field generating $p_t(x)$, and $\mathcal{U}[0, 1]$ denotes the uniform distribution on the unit interval $[0, 1]$. This loss is minimized if $v_\theta$ matches $u$, effectively following the probability

path from $p_0$ to $p_1$. FM offers several advantages over diffusion for neural network weight generation as it enables simpler and faster sampling, relies on direct vector field regression for training, and scales efficiently to high-dimensional spaces, making it particularly well-suited for generating complete neural network weights.

## 3.2 PERMUTATION SYMMETRIES OF NEURAL NETWORKS AND RE-BASIN

Permutation symmetry is a common weight space symmetry in neural networks (Hecht-Nielsen, 1990). Consider the activations $z_\ell \in \mathbb{R}^{d_\ell}$ at the $\ell^{\text{th}}$ layer of a simple MLP, with weights $W_\ell \in \mathbb{R}^{d_{\ell+1} \times d_\ell}$, biases $b_\ell \in \mathbb{R}^{d_{\ell+1}}$, and activation $\sigma$, $z_{\ell+1} = \sigma(W_\ell z_\ell + b_\ell)$, where $z_0 = x$ is the input data. Applying a permutation matrix $P \in \mathbb{R}^{d_{\ell+1} \times d_{\ell+1}}$ of appropriate dimension, yields

$$z_{\ell+1} = P^\mathsf{T} P z_{\ell+1} = P^\mathsf{T} P \sigma(W_\ell z_\ell + b_\ell) = P^\mathsf{T} \sigma(P W_\ell z_\ell + P b_\ell), \tag{2}$$

where $P^\mathsf{T} P = I$. This shows that a permutation of the output features of the $\ell^{th}$ layer, when met with the appropriate permutation of the input features of the next layer $\ell + 1$, will leave the overall MLP functionally invariant (Ainsworth et al., 2023).

Similar permutation symmetries (Lim et al., 2024) exist for the channels of convolutional neural networks and the attention heads of the transformer architecture (Hecht-Nielsen, 1990; Ainsworth et al., 2023; Rinaldi et al., 2025). These symmetries shape the loss landscape (Pittorino et al., 2022), impacting optimization(Neyshabur et al., 2015a; Liu, 2023; Zhao et al., 2024), generalization(Neyshabur et al., 2015b; Dinh et al., 2017), and model complexity (Zhao et al., 2025). They also impact the ability of generative models to learn distributions over neural network weights. Permutation symmetry gives rise to a highly multi-modal loss surface that renders the resulting models equivalent in task performance (Hecht-Nielsen, 1990; Lim et al., 2024).

In model alignment, weights are aligned with respect to a reference model to produce unique "canonical" representations for each equivalence class of the weight permutation symmetry. The Git Re-Basin (Ainsworth et al., 2023) weight matching approach permutes the hidden units of an MLP such that the inner product between reference and permuted weights is maximized. The resulting optimization problem is a sum of bilinear assignment problems (SOBLAP). Git Re-Basin solves this problem approximately, using coordinate descent, reducing each layer's subproblem to a linear assignment and iterating until convergence. TransFusion (Rinaldi et al., 2025) extends this idea of weight alignment to transformers where permutation symmetries exist both in MLPs and within and between attention heads, applying iterative alignment steps to reconcile permutations of heads and hidden units. More details on this can be found in Appendix A and Appendix B.

## 4 METHODS

We implement a simple MLP-based FM model. The explicit encoding of the symmetries of the neural networks is done using TransFusion for transformers and Git Re-Basin for all other architectures.

**Flow Matching Architecture and Training:** DeepWeightFlow uses a time-conditioned neural network that predicts a velocity vector along a trajectory between source and target network weights. The source is a distribution of Gaussian noise given by $x_0 \sim \mathcal{N}(0, \sigma^2 I)$, and the target is a distribution of trained weights ($x_1 \sim p_{\text{target}}$). The source distribution has the same dimensions as the target. Given a sampled time $t \in [0, 1]$ (uniformly distributed), an interpolated point along the straight-line trajectory is computed as $\mu_t = (1-t)x_0 + tx_1$. To stabilize training, stochastic points are generated by adding Gaussian noise $x_t = \mu_t + \epsilon$, with $\epsilon \sim \mathcal{N}(0, \sigma^2 I)$. The instantaneous target velocity along this linear trajectory is $u_t = x_1 - x_0$ (since $\frac{d\mu_t}{dt} = x_1 - x_0$), which is constant along the straight-line path. The network sees $x_t$ as input, while $u_t$ is derived from the endpoints $(x_0, x_1)$. The scalar time $t$ is embedded into a higher-dimensional vector $t_{\text{embed}} = \text{MLP}(t) \in \mathbb{R}^{d_{\text{time}}}$, where $d_{\text{time}}$ varies depending on the complexity of the model for which we are training DeepWeightFlow. We use a shallow MLP with layer normalization, dropout regularization, and GELU activations. This $t_{\text{embed}}$ is concatenated with $x_t$ and fed into the main network, allowing the network to condition on time in a learnable, flexible manner. The network maps $(x_t, t_{\text{embed}}) \mapsto v_\theta(x_t, t)$, where $v_\theta$ is the learned vector field. The main network consists of fully connected layers with LayerNorm, GELU activations, and Dropout, ending with a linear layer mapping back to the flattened weight dimension. Finally, new weight configurations are generated by integrating the learned vector field from

random Gaussian inputs in the same flattened weight space as the source distribution. This integration is performed using a fourth-order Runge-Kutta (RK4) method, which ensures high-accuracy trajectories in weight space. Concretely, at each integration step, the vector field is evaluated at the current point and time, and RK4 increments are computed to update the weights. This procedure allows sampling of realistic neural network weight configurations that smoothly interpolate between source and target distributions.

**Canonicalization:** *We apply canonicalization to align the training set to a single reference, as neural network loss landscapes are inherently degenerate due to permutation symmetries in the weight space.* This simplifies the learning process without the need for complex equivariant architectures. To implement canonicalization for smaller MLPs and ResNets, we use the weight-matching procedure of Git Re-Basin (Ainsworth et al., 2023) for 100 iterations. For ViTs, we use the TransFusion procedure (Rinaldi et al., 2025) for 10 iterations as the latter uses spectral decomposition and is slower than Git Re-Basin. The detailed description of these methods can be found in Appendix A and Appendix B. Subsection D.1 provides an estimate of the time required for canonicalization.

**Batch Normalization Statistics Based Recalibration:** *We implement a post-generation recalibration procedure where batch normalization (BN) (Ioffe & Szegedy, 2015) statistics are recomputed using the training dataset for each set of generated weights.* Neural networks with BN pose challenges for weight generation, as even perfectly generated weights can underperform if BN statistics are misaligned. DeepWeightFlow addresses this by recalibrating BN statistics after weight generation, ensuring models are accurate. While the FM framework successfully learns BN weight parameters ($\gamma$ and $\beta$), the running statistics (mean and variance) require more careful processing. These statistics are intrinsically tied to the training data distribution and must be precisely calibrated for each generated weight set. Our experiments, summarized in Table 7, reveal that directly transferring running statistics from a reference model yields suboptimal performance. We provide our recalibration algorithm in Algorithm 1 (Wortsman et al., 2021; 2022). Layer normalization (Ba et al., 2016) is permutation invariant and does not need recalibration (Ainsworth et al., 2023).

**Incremental and Dual PCA for scaling to large neural networks:** We use incremental and Dual PCA to scale to larger networks, as training on unprocessed training data for larger neural networks is limited by available GPU memory. We use incremental PCA to preprocess the training data when the weight space dimension is of $\mathcal{O}(10\text{M})$ and Dual PCA when the dimension of the weight space is $\mathcal{O}(100\text{M})$, and inverse PCA during generation. The algorithmic and computational details of the latter can be found in Subsection D.1. We also perform ablation studies to show the improvement in training time by using PCA (Table 8 in Appendix D).

**Training Data Generation:** All training data used in this work was generated *ab initio* from a set of randomly initialized neural networks trained separately, thus generating a diverse set of neural networks. Details of the training dataset generation can be found in Appendix E. We test DeepWeightFlow on diverse tasks such as the Iris (Fisher, 1936), MNIST (Lecun et al., 1998), Fashion-MNIST (Xiao et al., 2017), CIFAR-10 (Krizhevsky et al., 2009), and Yelp (Xiang Zhang, 2015) datasets for both classification and regression tasks. Recent work by Zeng et al. (2025) has raised concerns about the lack of diversity of weights sampled from generative models trained on checkpoints from training a single neural network (Wang et al., 2024). We generate neural network weights independently trained from random initialization and not drawn from a sequence of checkpoints from training a single neural network, thus increasing the diversity of the training set, for training all DeepWeightFlow models. We provide the hyperparameters in Appendix E. We provide code to generate the training dataset in `https://github.com/NNeuralDynamics/DeepWeightFlow` and hyperparameters in Table 12 and Table 13 of Appendix E. The datasets will be made available in the future.

## 5    EXPERIMENTS

We conduct a series of experiments to evaluate the effectiveness of our approach across different architectures, training conditions, and downstream tasks. We show that DeepWeightFlow generates complete weights for MLPs, ResNets, ViTs, and BERTs with high accuracy, and canonicalization improves performance at low FM model capacity. We see that incremental and Dual PCA enables scaling DeepWeightFlow to $\mathcal{O}(100\text{M})$ parameters. Our approach is robust across diverse initialization schemes, including Kaiming, Xavier, Gaussian, and Uniform. We see that Gaussian source

distributions outperform Kaiming, with variance choice being most critical at low capacity. Generated CIFAR-10 models transfer effectively to STL-10 and SVHN. Lastly, the generated neural networks are diverse while maintaining strong accuracy, and training and sampling are significantly faster than diffusion models such as RPG, D2NWG, and P-diff. Unless explicitly stated, all training sets are 100 terminal neural networks (not checkpoints from a single training round) initialized with unique seeds (Appendix E and Appendix F). All DeepWeightFlow models are architecture-specific except when we probe class-conditioning (Subsection K.2).

## 5.1 COMPLETE WEIGHT GENERATION ACROSS ARCHITECTURES

Table 1: *Comparison of DeepWeightFlow with other SOTA neural network weight generating methods for* **complete** *generation of weights for MNIST classifiers, without finetuning.*

| Model | Neural Network | Original | Generated | Reference |
|---|---|---|---|---|
| DeepWeightFlow (w/ Git Re-Basin) DeepWeightFlow (w/o Git Re-Basin) | 3-Layer MLP | $96.32 \pm 0.20$ | $96.17 \pm 0.31$ $96.19 \pm 0.27$ | |
| WeightFlow (Geometric, aligned + OT) | 3-Layer MLP | 93.3 | 78.6 | Erdogan (2025) |
| FLoWN (Unconditioned) | medium-CNN | 92.76 | 83.58 | Saragih et al. (2025b) |

Table 2: *Comparison of DeepWeightFlow with other SOTA neural network weight generating models for* **complete** *ResNet-18 CIFAR-10 classifier weight generation, without fine tuning.*

| Model | Original | Generated (Partial) | Generated (Complete) | Reference Reference |
|---|---|---|---|---|
| DeepWeightFlow (w/ Git Re-Basin) DeepWeightFlow (w/o Git Re-Basin) | $94.45 \pm 0.14$ | – – | $93.55 \pm 0.13$ $93.47 \pm 0.20$ | |
| RPG[†] | 95.3 | – | 95.1 | Wang et al. (2025) |
| SANE[†] | $92.14 \pm 0.12$ | – | $68.6 \pm 1.2$ | Schürholt et al. (2024) |
| D2NWG | 94.56 | $94.57 \pm 0.0$ | - | Soro et al. (2025) |
| N$\mathcal{M}$ (Unconditioned) | 94.54 | 94.36 | - | Saragih et al. (2025a) |
| P-diff (best neural network) | 94.54 | 94.36 | – | Wang et al. (2024) (Saragih et al., 2025b) |
| FLoWN (best neural network) | 94.54 | 94.36 | – | Saragih et al. (2025b) |

[†]Models use autoregression to generate *complete* models over multiple passes.

Table 3: *Comparison of DeepWeightFlow with other SOTA neural network weight generating models for* **complete** *ResNet-18 STL-10 classifier weight generation, without fine-tuning.*

| Model | Original | Generated (Partial) | Generated (Complete) | Reference |
|---|---|---|---|---|
| DeepWeightFlow (w/ Re-Basin) DeepWeightFlow (w/o Re-Basin) | $62.30 \pm 0.77$ | – – | $62.46 \pm 0.79$ $62.50 \pm 0.66$ | |
| P-diff | 62.00 | 62.24 | – | Wang et al. (2024) |
| FLoWN | 62.00 | 62.00 | – | Saragih et al. (2025b) |
| N$\mathcal{M}$ (Unconditioned) | 62.00 | 62.00 | – | Saragih et al. (2025a) |

Table 4: *Comparison of DeepWeightFlow with other SOTA neural network weight generating models for ViT family CIFAR-10 classifiers, without finetuning. We have used ViT-small-192, indicating an embedding dimension of 192* Wang et al. (2025)*;* Schürholt et al. (2024)*;* Soro et al. (2025)*;* Dosovitskiy et al. (2021)*.*

| Model | Neural Network | Original | Generated | Reference |
|---|---|---|---|---|
| DeepWeightFlow (w/ TransFusion) DeepWeightFlow (w/o TransFusion) | Vit-Small-192 | $83.30 \pm 0.29$ | $83.07 \pm 0.42$ $82.58 \pm 0.07$ | |
| P-diff (Best) | ViT-mini | 73.0 | 73.6 | Wang et al. (2024) |
| RPG | ViT-Base | 98.7 | 98.9 | Wang et al. (2025) |

*DeepWeightFlow generates complete neural network weights and the generated networks perform as well as the training set.* In Table 1, Table 2, Table 3, and Table 4, we highlight the results of generating MLPs, ResNet-18/20s and ViTs from DeepWeightFlow models. We have conducted our experiments on MNIST, Fashion-MNIST, CIFAR-10, STL-10 (Coates et al., 2011), and SVNH (Goodfel-

low et al., 2013) datasets. As noted before, we generate the *complete* weights for all neural networks, including those with batch normalization such as ResNet-18 and ResNet-20. The comprehensive weight generation scope of DeepWeightFlow is unlike existing approaches such as FLoWN (Saragih et al., 2025b) and P-diff (Wang et al., 2024), which primarily generate only partial weight sets (limited to batch normalization parameters due to lack of scalability with neural network parameter size). Moreover, DeepWeightFlow generated networks perform as well as the training set without the requirement of additional conditioning during training or inference. With sufficient flow model capacity, performance converges regardless of canonicalization or noise scheduling strategy, suggesting that model capacity can compensate for suboptimal design choices. The choice of source distribution significantly impacts FM performance and generated model diversity (cf. Figure 2).

**Effect of Source Distributions:** *Critical to the success of DeepWeightFlow, is the careful selection of the standard deviation parameter of the source distribution: optimal results are achieved when the source distribution's standard deviation matches or slightly undershoots that of the target weight distribution.* Our empirical analysis demonstrates that Gaussian noise consistently outperforms alternative initializations (e.g., Kaiming initialization (He et al., 2015)) as the source distribution (Table 16 in Appendix H). This sensitivity is particularly pronounced in smaller flow models, where insufficient capacity amplifies the importance of proper initialization (Saragih et al., 2025b).

Table 5: *Canonicalization is beneficial when DeepWeightFlow has limited capacity, leading to superior performance. As model capacity increases, both canonicalized and non-canonicalized models perform comparably, with the best results highlighted in bold.*

| Dataset | Architecture | $d_h$ * | Original | Generated | |
|---|---|---|---|---|---|
| | | | (metric) mean $\pm$ st. dev. | with Re-Basin (metric) mean $\pm$ st. dev. | without Re-Basin (metric) mean $\pm$ st. dev. |
| *Classification Tasks (Accuracy %)* | | | | | |
| Iris | MLP | 256 128 64 32 | 90.70 $\pm$ 2.02 | **91.43 $\pm$ 2.07** **91.43 $\pm$ 2.46** **91.87 $\pm$ 2.23** **90.80 $\pm$ 2.54** | 91.03 $\pm$ 2.20 90.87 $\pm$ 3.25 90.80 $\pm$ 4.86 88.93 $\pm$ 6.09 |
| MNIST | MLP | 512 256 128 64 | 96.32 $\pm$ 0.20 | 96.17 $\pm$ 0.31 **96.21 $\pm$ 0.28** **91.74 $\pm$ 10.37** **57.80 $\pm$ 9.85** | **96.19 $\pm$ 0.27** 96.20 $\pm$ 0.23 89.71 $\pm$ 17.93 25.54 $\pm$ 12.90 |
| Fashion-MNIST | MLP | 512 256 128 64 | 89.24 $\pm$ 0.27 | 89.10 $\pm$ 0.29 **89.06 $\pm$ 0.29** **88.09 $\pm$ 2.24** **77.76 $\pm$ 3.72** | **89.11 $\pm$ 0.28** 89.02 $\pm$ 0.30 85.81 $\pm$ 11.32 53.35 $\pm$ 30.49 |
| CIFAR-10 | ResNet-20 | 512 256 128 64 | 73.62 $\pm$ 2.24 | **75.07 $\pm$ 1.24** **75.32 $\pm$ 0.83** **73.08 $\pm$ 4.35** **20.16 $\pm$ 13.44** | 74.92 $\pm$ 0.80 74.91 $\pm$ 0.97 72.35 $\pm$ 8.86 20.06 $\pm$ 15.76 |
| CIFAR-10 | Vit-Small-192 | 384 256 128 64 | 83.30 $\pm$ 0.29 | **82.99 $\pm$ 0.11** **83.07 $\pm$ 0.42** **69.09 $\pm$ 25.20** **43.13 $\pm$ 30.28** | 82.58 $\pm$ 0.07 82.51 $\pm$ 0.55 41.15 $\pm$ 25.26 12.67 $\pm$ 7.11 |
| CIFAR-10 | ResNet-18[†] | 1024 512 128 64 | 94.45 $\pm$ 0.14 | **93.55 $\pm$ 0.13** **93.49 $\pm$ 0.19** **57.98 $\pm$ 34.02** **29.92 $\pm$ 19.79** | 93.47 $\pm$ 0.20 93.43 $\pm$ 0.64 47.55 $\pm$ 37.46 21.93 $\pm$ 19.86 |
| *Regression Task (Spearman Correlation)* | | | | | |
| Yelp Review | BERT-118M[‡] | 1024 768 | 0.7902 $\pm$ 0.061 | **0.7909 $\pm$ 0.005** **0.7894 $\pm$ 0.006** | 0.7884 $\pm$ 0.012 0.7892 $\pm$ 0.015 |

[†]ResNet-18 results use standard incremental PCA-reduced weights.
[‡]BERT-118M results use dual/Gram PCA approach.
*$d_h$: flow hidden dimension

**Scaling with PCA:** *DeepWeightFlow can scale to large neural networks using PCA* (Wold et al., 1987; Hotelling, 1933). For models with tens of millions of parameters, we employ incremental PCA (Ross et al., 2008) to reduce the dimensionality of flattened weight vectors in the training set, and inverse transformation post-generation. This approach maintains accuracy levels, as can be seen from Table 8 in Appendix D, while enabling tractable training of DeepWeightFlow for large-scale architectures. This demonstrates the feasibility of extending our methodology to generate complete weight sets for contemporary large neural networks without the requirement of training additional models for dimensionality reduction, such as autoencoders, as is often done for latent diffusion-

based models (Wang et al., 2024). We demonstrate that DeepWeightFlow can be scaled to $\mathcal{O}(100M)$ parameters with Dual PCA. Given the reduction of resources and time required with Dual PCA, we estimate that models of $\mathcal{O}(1B)$ parameters might be possible to generate using DeepWeightFlow and leave that as future work.

**Impact of Canonicalization:** *We observe a capacity-dependent behavior of DeepWeightFlow models with and without canonicalization.* At lower capacity of the FM models, models trained on canonicalized neural network weights generate higher performing ensembles than the FM models trained on non-canonicalized data. However, as the capacity of the FM model increases, the performance of the ensembles of generated neural networks become similar. In general, FM models trained on canonicalized neural network weights approach the performance of the training set ("original" neural networks) with lower capacity. Moreover, when flow model parameters are limited, models trained on canonicalized data generate neural networks with observably lower variance in accuracy compared to non-canonicalized counterparts. In Table 5, we show the performance of DeepWeightFlow with and without canonicalization.

**Robustness Across Initialization Schemes:** To evaluate generalization capability, we conducted extensive robustness testing using MLP models trained on the Iris dataset with diverse initialization strategies (Kaiming (He et al., 2015), Xavier (Glorot & Bengio, 2010), Kaiming weights and zero for biases, normal, and uniform distributions). *Training a single flow model on this heterogeneous collection (100 models total: 20 seeds × 5 initialization types) successfully generated novel weights achieving high test accuracy, demonstrating the framework's ability to learn from and generate weights across different initialization regimes.* All other experiments maintained consistency by using Kaiming initialization with varied random seeds.

## 5.2 TRANSFER LEARNING ON UNSEEN DATASETS

*Our generated models can be effectively used for transfer learning (Nava et al., 2023; Zhang et al., 2024) across unseen datasets.* In our experiments, we trained DeepWeightFlow on ResNet-18 models for the CIFAR-10 dataset using PCA, generated 5 models, and recalibrated their batch normalization running mean and variance on a small subset of CIFAR-10 in the same way as applied in Table 5 and elaborated on in Table 14. These models were then evaluated under zero-shot and finetuning settings on STL-10 and SVHN datasets. The results are presented in Table 6. DeepWeightFlow-generated models consistently outperformed state-of-the-art FM models such as FloWN (Saragih et al., 2025b) in both zero-shot and finetuning evaluations. Furthermore, they significantly outperformed randomly initialized models, proving the effectiveness of the method. The same comparison is done with SANE (Schürholt et al., 2024) and reaches the same conclusion. Results on transfer learning for CIFAR-100 models fine-tuned on CIFAR-10 ResNet-18 backbone can be found in Appendix J.

## 5.3 DIVERSITY OF GENERATED MODELS

To evaluate the DeepWeightFlow models' generative capabilities, we compute the maximum IoU (mIoU) between the generated neural networks and the neural networks in the training set (referred to as the "original" neural networks). The mIoU is constructed from the intersection over union of the wrong predictions made by the neural networks (Wang et al., 2024). It is defined as $\text{IoU} = |P_1^{\text{wrong}} \cap P_2^{\text{wrong}}|/|P_1^{\text{wrong}} \cup P_2^{\text{wrong}}|$. where $P_1$ comes from the set being compared (such as from the generated set) and $P_2$ comes from a reference set (such as the set of original neural networks). We disregard the IoU of a neural network with itself as it is trivially 1. The mIoU measure scales from complete dissimilarity at 0 to complete similarity at 1.

In Figure 2, we compare the original neural networks with the generated ones, with noise added to the weights of the original neural networks, and with neural networks generated with different FM source distributions. The upper row compares the cases for the FM models trained with Re-Basin, and the lower panels, without. In the *left-most panels*, we see that i) the original networks are quite diverse from each other, as evident from the blue cloud. This is the case as, unlike several previous works, we do not use checkpoints from the training of a single neural network as the training set of the DeepWeightFlow model. The training set for DeepWeightFlow consists strictly of terminal models of unique random initializations. Details for dataset generation are outlined in Appendix E. ii) The yellow and green clouds show that adding progressively increasing Gaussian noise to the

Table 6: *Transfer learning performance across different architectures. For ResNet-18, we compare CIFAR-10 classifiers generated by DeepWeightFlow, FLoWN, and RandomInit. For SmallCNN, we compare with SANE (Schürholt et al., 2024) trained on CIFAR-10 and transferred to STL-10 using the same architecture as mentioned in Schürholt et al. (2024). RandomInit refers to randomly initialized neural networks with Kaiming-He initialization. Pretrained refers to neural networks from our training dataset, and Generated refers to weights sampled from the respective generative model.*

| Architecture | Epoch | Model | Method | STL-10 | SVHN |
|---|---|---|---|---|---|
| *ResNet-18 Results (Comparison with FLoWN (Saragih et al., 2025b))* | | | | | |
| ResNet-18 | 0 | FLoWN | RandomInit | $10.00 \pm 0.00$ | $10.00 \pm 0.00$ |
| | | | Generated | $35.16 \pm 1.24$ | $\mathbf{17.99 \pm 0.82}$ |
| | | DeepWeightFlow | RandomInit | $11.18 \pm 1.48$ | $8.01 \pm 1.41$ |
| | | | Pretrained | $48.31 \pm 0.17$ | $11.51 \pm 0.31$ |
| | | | Generated | $\mathbf{48.32 \pm 0.34}$ | $11.57 \pm 0.49$ |
| ResNet-18 | 1 | FLoWN | RandomInit | $18.94 \pm 0.09$ | $19.50 \pm 0.03$ |
| | | | Generated | $36.15 \pm 1.14$ | $68.64 \pm 7.07$ |
| | | DeepWeightFlow | RandomInit | $38.28 \pm 1.07$ | $84.07 \pm 1.76$ |
| | | | Pretrained | $\mathbf{79.81 \pm 0.54}$ | $91.29 \pm 0.76$ |
| | | | Generated | $79.69 \pm 1.08$ | $\mathbf{91.66 \pm 0.79}$ |
| ResNet-18 | 5 | FLoWN | RandomInit | $28.24 \pm 0.01$ | $39.59 \pm 10.0$ |
| | | | Generated | $37.43 \pm 1.19$ | $77.36 \pm 1.07$ |
| | | DeepWeightFlow | RandomInit | $51.35 \pm 0.51$ | $93.82 \pm 0.16$ |
| | | | Pretrained | $84.61 \pm 0.21$ | $95.82 \pm 0.16$ |
| | | | Generated | $\mathbf{84.63 \pm 0.17}$ | $\mathbf{95.85 \pm 0.09}$ |
| *SmallCNN Results (Comparison with SANE (Schürholt et al., 2024))* | | | | | |
| SmallCNN | 0 | SANE | Train fr. scratch | $\sim 10$ | – |
| | | | Pretrained | $16.2 \pm 2.3$ | – |
| | | | $SANE_{SUB}$ | $17.4 \pm 1.4$ | – |
| | | DeepWeightFlow | RandomInit | $9.47 \pm 0.52$ | – |
| | | | Pretrained | $35.18 \pm 0.71$ | – |
| | | | Generated | $\mathbf{35.29 \pm 0.48}$ | – |
| SmallCNN | 1 | SANE | Train fr. scratch | $21.3 \pm 1.6$ | – |
| | | | Pretrained | $24.8 \pm 0.8$ | – |
| | | | $SANE_{SUB}$ | $25.6 \pm 1.7$ | – |
| | | DeepWeightFlow | RandomInit | $21.09 \pm 2.52$ | – |
| | | | Pretrained | $\mathbf{41.66 \pm 1.75}$ | – |
| | | | Generated | $41.03 \pm 1.22$ | – |
| SmallCNN | 25 | SANE | Train fr. scratch | $44.0 \pm 1.0$ | – |
| | | | Pretrained | $49.0 \pm 0.9$ | – |
| | | | $SANE_{SUB}$ | $49.8 \pm 0.6$ | – |
| | | DeepWeightFlow | RandomInit | $44.33 \pm 1.54$ | – |
| | | | Pretrained | $62.14 \pm 0.84$ | – |
| | | | Generated | $\mathbf{62.62 \pm 0.46}$ | – |

original networks makes them progressively diverse from the original networks as expected (IoU < 1). iii) The red cloud representing the generated networks shows diversity from the original set but seems to overlap with the green set, which represents the set created by adding noise sampled from $\mathcal{N}(0, 0.01)$ to the original neural network weights.

From the middle panels in Figure 2, we see that the red cloud representing the generated neural networks is sufficiently diverse from the original ones with added noise sampled from $\mathcal{N}(0, 0.01)$. This gives us confidence that the generated neural networks are, indeed, *not the same* as the original networks with noise added to the weights. Lastly, the right-most panels show how diverse the generated neural networks are when generated with different source distributions. Hence, DeepWeightFlow is capable of generating a diverse set of neural networks while maintaining the accuracy of the task. In Appendix I, we provide the numerical estimates of mIoU, the Jensen-Shannon, Wasserstein, $L^2$, cosine similarity, and Nearest Neighbors (NN) distances between generated and original neural networks and supplemental mIoU analysis of ResNet-18 weights generated by DeepWeightFlow.

## 5.4 TRAINING AND SAMPLING EFFICIENCY

*DeepWeightFlow is significantly faster to train and generate neural network weights when compared to diffusion models in complete neural network weights generation.* DeepWeightFlow takes up to $\mathcal{O}(10)$ minutes to train for most neural network architectures with up to $\mathcal{O}(100M)$ parameters. as compared to the several hours that it takes to train RPG (Wang et al., 2025). DeepWeightFlow takes a few seconds to generate neural networks compared to the minutes or hours it takes to generate

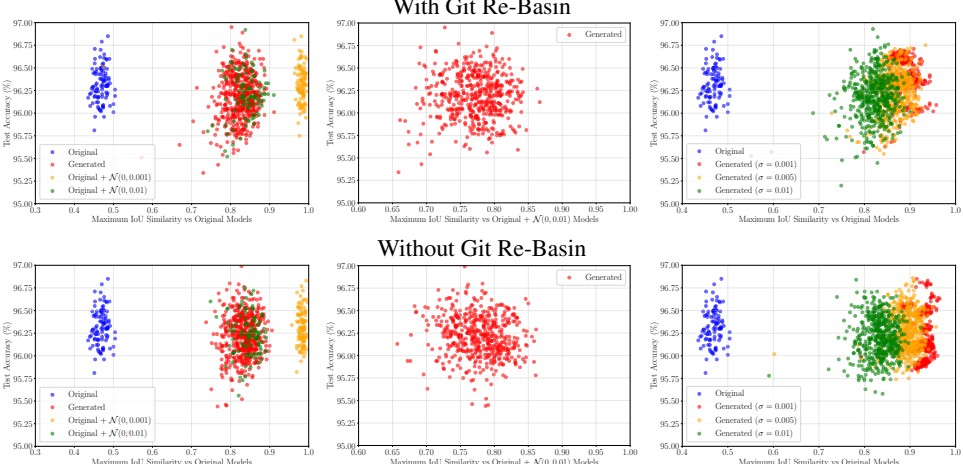

Figure 2: *Maximum IoU vs test set accuracy for MNIST classifying MLPs. Lower maximum IoU implies greater diversity in the neural network weights. The left panels are generated and original neural networks (from the DeepWeightFlow training set) with different scales of Gaussian noise added to the original neural networks. The middle panels show that the generated neural networks and the original neural networks with noise added, which overlap in the left panels, are concretely different. The right panels contain the original and generated neural networks with different source distributions. All panels include 500 generated neural networks.*

using RPG, P-Diff, or D2NWG. Yet, DeepWeightFlow generates ensembles of neural networks that have comparable outcomes for ResNet-18s and ViTs. This is primarily because the other models are diffusion models, whereas DeepWeightFlow is based on FM using a simple MLP implementation. A detailed comparison of training and generation efficiency can be found in Appendix G.

## 6 CONCLUSION

In this work, we introduce DeepWeightFlow, a generative model for neural network weights that performs FM *directly* in weight space, unconditioned by dataset characteristics, task descriptions, or architectural specifications, and avoiding nonlinear dimensionality reduction. We show that Deep-WeightFlow generates diverse neural network weights for a variety of architectures (MLP, ResNet, ViT, BERT) that show excellent performance on vision, tabular classification, and natural language tasks (regression). We provide empirical evidence that canonicalizing the training data facilitates the generation of larger networks but is of limited use for moderate-dimensional weights or with increasing FM model capacity. DeepWeightFlow can be combined with simple linear dimensionality reduction techniques like incremental PCA and Dual PCA to alleviate restrictions on neural network size and demonstrate scalability to large neural networks of $\mathcal{O}(100M)$ parameters with possibilities of scaling even further. The compatibility of DeepWeightFlow with model distillation, low-rank approximations, or sparsity remains as future work. As such, some open questions about the relative merits of canonicalization, equivariant architecture design, and data augmentation for learning in deep weight spaces remain. Lastly, we demonstrate DeepWeightFlow's ability to generalize to multi-class generation through class conditioning (Appendix K). We extend Deep-WeightFlow to combining multi-class and multi-architecture generation of complete weights. The results do not seem promising and we leave further exploration to future work with possibilities of combining DeepWeightFlow and dataset conditioning similar to FLoWN or D2NWG. Nevertheless, DeepWeightFlow shows promise for extension to real-world applications such as rapid generation of neural networks for vision and NLP tasks in distributed devices for sensing of changing environments and in privacy-protecting model distribution to avoid leakage of training data.

### REPRODUCIBILITY STATEMENT

The architectural details along with the hyperparameters used to generate the data have been provided in the main text and Appendix E and Appendix F. The dataset will be made available on request and/or uploaded to a data repository. The code necessary to reproduce the results is in https://github.com/NNeuralDynamics/DeepWeightFlow.

AUTHOR CONTRIBUTIONS

S.G. and S.B. contributed to all aspects of the work including ideation, implementation, analysis and drafts. M.L. and Z.S. contributed to ideation, analysis, and drafting of the work. R.W. and A.P. contributed to the ideation, methodological guidance, and drafting of the work.

ACKNOWLEDGEMENTS

R.W. would like to acknowledge support from NSF Grants 2442658 and 2134178. This work is supported by the National Science Foundation under Cooperative Agreement PHY-2019786 (The NSF AI Institute for Artificial Intelligence and Fundamental Interactions, http://iaifi.org/). M.L. was supported by the Joseph E. Aoun Endowment. S.G. and M.L. would also like to acknowledge Derek Lim for a fruitful discussion about this work. The authors are grateful to Voltage Park for providing computational resources that enabled part of the work.

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

## A   GIT RE-BASIN

Git Re-Basin weight matching, formulated by Ainsworth et al. (2023), is a greedy permutation coordinate descent algorithm for moving a model's weights $\theta_A$ into the same 'basin' in the loss landscape of the model class $f_{\hat{\theta}}$ as a reference model's weights $\theta_B$.

This operation is applied here as a canonicalization step before weight flattening and the subsequent training of the DeepWeightFlow models. The procedure reduces the space of the task from $\mathbb{R}^\theta$ to a quotient space of $\mathbb{R}^\theta$ modulo permutation symmetry.

Applying this across the model layers constructs a transformed model $\theta'$ by

$$W'_\ell = PW_\ell,\ b'_\ell = Pb_\ell,\ W'_{\ell+1} = W_{\ell+1}P^T \tag{3}$$

The 'distance' between two permutations is therefore a Frobenius inner product of $P_\ell W_\ell^A$ and $W_\ell^B$, written as $\langle A, B \rangle = \sum_{i,j} A_{i,j} B_{i,j}$ for real-valued matrices $A$ and $B$. Accounting for the transforms outlined above, the process of matching the permutations across the stack of layers becomes,

$$\arg\max_{\pi=\{P_\ell\}_1^L} \sum_{n=1}^L \left\langle W_i^B, P_i W_i^A P_{i-1}^T \right\rangle \text{ with } P_0^T = I \tag{4}$$

This formulation presents a Symmetric Orthogonal Bilinear Assignment Problem (SOBLAP), which is NP-hard. However, when relaxed to focus on a single permutation $P_\ell$ at a time - *ceteris paribus*, the problem simplifies to a series of Linear Assignment Problems (LAPs) of the form below (Ainsworth et al., 2023; Zhao et al., 2025; Rinaldi et al., 2025). These LAPs can be solved in polynomial time by methods like the Hungarian Algorithm (Jonker & Volgenant, 1987).

$$\arg\max_{P_\ell} \left\langle W_\ell^B, P_\ell W_\ell^A P_{\ell-1}^T \right\rangle + \left\langle W_{\ell+1}^B, P_{\ell+1} W_{\ell+1}^A P_\ell^T \right\rangle \tag{5}$$

The product of this process is a permutation $\pi'$ of model $A$'s weights into the same basin in $f_\theta$'s loss landscape as model $B$ with exact functional equivalence ($f_{\theta_A} = f_{\pi'(\theta_A)}$). However, sequences of LAPs are understood to be coarse approximations of SOBLAPs and, as such, strong conclusions cannot be drawn about the optimality of $\pi'$ (Rinaldi et al., 2025; Ainsworth et al., 2023).

## B   TRANSFUSION

We canonicalize a collection of Vision Transformers (ViTs) using the method of Rinaldi et al. (2025), which introduces a structured alignment procedure for multi-head attention transformer weights (Rinaldi et al., 2025).

The core difficulty in transformers arises from multi-head attention and residual connections: Naive global permutations either mix information across heads or break functional equivalence in residual branches (Zhao et al., 2025). To address this, the method applies a *two-level permutation scheme*:

1. **Inter-Head Alignment:** For each multi-head attention layer, attention heads from different checkpoints are first matched. This is done by comparing the singular value spectra of their projection matrices, which are invariant under row and column permutations, and then solving the resulting assignment problem with the Hungarian algorithm. This step ensures that corresponding heads are correctly paired across models.

   For a sub matrix representing a single attention head in model $A$, $h_i^A = [\tilde{W}]_i^A \in \mathbb{R}^{k \times m}$, where $k$ is the key value dimension and $m$ is the attention embedding dimension, apply

singular value decomposition ($A = U\Sigma V^T$) to access the spectral projection matricies $\Sigma$, which are invariant to row and column permutations. For every head in a layer of model $A$, construct a distance, $d_i, j = ||\Sigma_i - \Sigma_j||$. These distances can be constructed for $q$, $k$, and $v$ for each head and combined linearly $D_{i,j} = d_{i,j}^q + d_{i,j}^k + d_{i,j}^v$ with $D_{i,j} \in \mathbb{R}^{H \times H}$ ($H$ is the number of heads). Therefore the optimal pairing of heads for model $A$ and $B$ is (Rinaldi et al., 2025),

$$P_{\text{inter head}} = \arg\min_{P \in S_H} \sum D_{i,P[i]} \tag{6}$$

2. **Intra-Head Alignment:** Once heads are paired, the method refines the alignment by permuting rows and columns *within* each head independently, again solved via assignment on pairwise similarity scores. Restricting permutations within heads preserves head isolation and guarantees that residual connections remain valid after alignment.

   After matching the heads of $A$ to $B$ the goal aligns closely with Git Re-Basin (Ainsworth et al., 2023) - to reorder $h_{P[i]}^A$ such that the Frobenius inner product is maximized between $H$ sub portions (Rinaldi et al., 2025),

$$P_{\text{intra head}}^{(i)} = \arg\max \langle h_i^B, P h_{P[i]}^A \rangle \tag{7}$$

By iterating these two stages across all transformer layers, the procedure yields a canonicalized parameterization in which weights are aligned up to permutation symmetries. The goal is to permute units in such a way that two weight sets $\theta_A$ and $\theta_B$ become functionally comparable, reducing the effective size of the weight space that the FM encounters Rinaldi et al. (2025). This is similar to the case of Git Re-Basin (Ainsworth et al., 2023) for canonicalization.

## C  RECALIBRATION OF BATCH NORMALIZATION WEIGHTS

Given a generated neural network with randomly initialized or flow-matched weights, the batch normalization layers contain statistics that may not match the actual data distribution. Naively interpolating weights of trained networks can lead to variance collapse (Jordan et al., 2022; Ainsworth et al., 2023), where the per-channel activation variances shrink drastically, breaking normalization and degrading performance. The recalibration process computes proper running statistics using the target dataset(Izmailov et al., 2018; Maddox et al., 2019; Shomron & Weiser, 2020; Wang et al., 2021).

We include these statistics parameters of batch normalization layers in the PermutationSpec of Git Re-Basin, a config that defines the permutation ordering across layers for weight matching, so that these statistics are also permuted and correctly maintained, ensuring that the permuted networks retain the same weights and accuracy as the original network.

### C.1  STANDARD BATCH NORMALIZATION

For a feature map $\mathbf{x} \in \mathbb{R}^{N \times C \times H \times W}$ where $N$ is batch size, $C$ is channels, and $H, W$ are spatial dimensions:

$$\mu_c = \frac{1}{NHW} \sum_{n=1}^{N} \sum_{h=1}^{H} \sum_{w=1}^{W} x_{n,c,h,w} \tag{8}$$

$$\sigma_c^2 = \frac{1}{NHW} \sum_{n=1}^{N} \sum_{h=1}^{H} \sum_{w=1}^{W} (x_{n,c,h,w} - \mu_c)^2 \tag{9}$$

$$\hat{x}_{n,c,h,w} = \frac{x_{n,c,h,w} - \mu_c}{\sqrt{\sigma_c^2 + \epsilon}} \tag{10}$$

$$y_{n,c,h,w} = \gamma_c \hat{x}_{n,c,h,w} + \beta_c \tag{11}$$

where $\gamma_c$ and $\beta_c$ are learnable scale and shift parameters, and $\epsilon$ is a small constant for numerical stability. During training, BatchNorm (Ioffe & Szegedy, 2015) maintains running statistics using

an exponential moving average:

$$\bar{\mu}_c^{(t)} = (1 - \alpha)\bar{\mu}_c^{(t-1)} + \alpha\mu_c^{(t)} \tag{12}$$

$$\bar{\sigma}_c^{2(t)} = (1 - \alpha)\bar{\sigma}_c^{2(t-1)} + \alpha\sigma_c^{2(t)} \tag{13}$$

where $\alpha$ is the momentum parameter, typically $0.1$, and $t$ denotes the time step.

---

**Algorithm 1** Batch Normalization Recalibration

---

1: **Input:** Calibration dataset $\mathcal{D}$ (e.g., test dataset), batch size $B$
2: $H$ and $W$ denote the height and width of feature maps
3: $x_{i,c,h,w}$ denotes the activation of sample $i$, channel $c$, at spatial position $(h, w)$.
4: Initialize $\bar{\mu}_c = 0$, $\bar{\sigma}_c^2 = 1$, $n_c = 0$ for all channels $c$
5: Disable exponential moving average (momentum) updates
6: Partition $\mathcal{D}$ into mini-batch sequence $\{\mathcal{B}_1, \mathcal{B}_2, \ldots, \mathcal{B}_K\}$ where $\bigcup_{k=1}^{K} \mathcal{B}_k = \mathcal{D}$
7: Define batch statistics for each $\mathcal{B}_k$ and channel $c$:

$$\mu_c^{(k)} = \frac{1}{|\mathcal{B}_k|HW} \sum_{i \in \mathcal{B}_k} \sum_{h=1}^{H} \sum_{w=1}^{W} x_{i,c,h,w}$$

$$\sigma_c^{2(k)} = \frac{1}{|\mathcal{B}_k|HW} \sum_{i \in \mathcal{B}_k} \sum_{h=1}^{H} \sum_{w=1}^{W} (x_{i,c,h,w} - \mu_c^{(k)})^2$$

8: Compute running statistics where $n_k = |\mathcal{B}_k|HW$ and $n_c^{(k)} = n_c^{(k-1)} + n_k$:

$$\bar{\mu}_c^{(k)} = \frac{n_c^{(k-1)}\bar{\mu}_c^{(k-1)} + n_k \cdot \mu_c^{(k)}}{n_c^{(k)}}$$

$$\bar{\sigma}_c^{2(k)} = \frac{n_c^{(k-1)}\bar{\sigma}_c^{2(k-1)} + n_k \cdot \sigma_c^{2(k)} + \frac{n_c^{(k-1)}n_k}{n_c^{(k)}}\left(\bar{\mu}_c^{(k-1)} - \mu_c^{(k)}\right)^2}{n_c^{(k)}}$$

9: Final recalibrated statistics: $\bar{\mu}_c = \bar{\mu}_c^{(K)}$, $\bar{\sigma}_c^2 = \bar{\sigma}_c^{2(K)}$ for all channels $c$
10: Restore exponential moving average updates (set momentum = 0.1)

---

### C.2 RECALIBRATION PROCESS

For generated networks, recompute running BatchNorm statistics:

1. **Reset**: Initialize running mean and variance for all channels, and set total sample count to zero.

2. **Disable momentum**: Turn off exponential moving average updates.

3. **Forward pass and incremental update**: For each mini-batch in the calibration dataset:
   - Compute the mean and variance of the batch for each channel.
   - Update the running mean as a weighted average of the previous running mean and the batch mean.
   - Update the running variance by combining the previous variance, the batch variance, and a correction for the shift in means.
   - Update the total sample count.

4. **Restore momentum**: Re-enable exponential moving average updates with the original momentum value.

The algorithm we use for recalibration of the batch normalization running statistics is provided in Algorithm 1. In Table 7 we show the results of recalibration on the generated neural networks. This clearly shows the importance of batch normalization, running statistics recalibration on the generation of neural networks that have batch normalization in their architecture.

Table 7: *Comparing the impact of batch norm recalibration on complete ResNet-18 and 20s generated by DeepWeightFlow. Recalibrating batch normalization statistics on a small subset of target data significantly improves the accuracy of generated models.*

| Model | Git Re-Basin | Strategy | Mean ± Std (%) | Min (%) | Max (%) |
|---|---|---|---|---|---|
| ResNet-18 | Yes | No Calibration | 10.00 ± 0.00 | 10.00 | 10.00 |
|  |  | Ref BN* | 19.06 ± 9.68 | 10.00 | 94.05 |
|  |  | Recalibrated | **93.05 ± 4.42** | 49.12 | 93.93 |
| ResNet-18 | No | No Calibration | 10.00 ± 0.00 | 10.00 | 10.00 |
|  |  | Ref BN | 10.28 ± 1.24 | 6.23 | 15.93 |
|  |  | Recalibrated | **93.49 ± 0.21** | 92.77 | 93.96 |
| ResNet-20 | Yes | No Calibration | 14.36 ± 3.10 | 5.84 | 19.03 |
|  |  | Ref BN | 17.88 ± 4.66 | 9.96 | 26.54 |
|  |  | Recalibrated | **74.57 ± 0.84** | 71.47 | 76.17 |
| ResNet-20 | No | No Calibration | 12.64 ± 2.22 | 8.12 | 18.19 |
|  |  | Ref BN | 10.23 ± 0.79 | 8.04 | 14.92 |
|  |  | Recalibrated | **75.21 ± 0.79** | 72.06 | 76.52 |

\* Ref BN: Uses batch normalization statistics from reference model (seed 0)

# D  PCA AS AN EFFECTIVE COMPRESSION STRATEGY

Table 8: *Accuracy and efficiency comparison of DeepWeightFlow with and without incremental PCA compression. Training/generation times in minutes. Generation time is the total generation+ inference time for 100 models.*

| Model | Method | $d_h$ | Original Mean | Generated (Accuracy) With Re-basin | Generated (Accuracy) Without Re-basin | Time (min) Train | Time (min) Generation |
|---|---|---|---|---|---|---|---|
| ResNet-20 | Without PCA | 512 | 73.62 ± 2.24 | 75.07 ± 1.24 | 74.92 ± 0.80 | 11.25 | 6.00 |
| ResNet-20 | **With PCA** | 512 | 73.62 ± 2.24 | **75.96 ± 0.89** | **75.97 ± 0.86** | **1.23** | **5.78** |
| Vit-Small-192 | Without PCA | 384 | 83.30 ± 0.29 | 82.99 ± 0.11 | 82.58 ± 0.07 | 21.00 | 3.60 |
| Vit-Small-192 | **With PCA** | 1024 | 83.30 ± 0.29 | **83.08 ± 0.19** | **83.28 ± 0.01** | **2.90** | **1.75** |

In Table 8, we show the effects of using PCA to reduce the dimension of the neural network weight space. This is necessary as DeepWeightFlow cannot be trained on with the full rank of the larger neural networks, such as ResNet-18, due to memory constraints on a single GPU. Hence, we reduce dimensionality using PCA and decompress after generation. To test the validity of PCA, we trained the DeepWeightFlow models on ResNet-20 and ViT with and without using PCA as shown in Table 8. We observe that the accuracy and diversity of the neural networks (indicated by the standard deviation in the accuracy) are sufficiently representative of the original sample with or without PCA. This gives us confidence that much larger neural networks can be generated by DeepWeightFlow using PCA. We leave the complete implementation of this as future work.

Here we have performed incremental PCA that lets us perform PCA in chunks without loading all data into memory, but the math and essential foundation for it is exactly the same as standard PCA. Incremental PCA reduces the dimensionality of the generated weight matrices, we start with data of shape $(n_{\text{samples}}, \text{flat\_dim})$, incremental PCA projects it into a latent space of size $(n_{\text{samples}}, \text{latent\_dim})$, where we set $\text{latent\_dim} = 99$. Since PCA orders components by explained variance and the rank of the data matrix is bounded by $n_{\text{samples}} - 1$, at most 99 meaningful directions can exist for 100 samples we used. Therefore, using 99 principal components retains essentially all the variance of the dataset, while compressing the original high-dimensional representation into a very compact latent space.

## D.1  DUAL PCA

While we have demonstrated results using incremental PCA for models with tens of millions of parameters, scaling to models with up to 100M parameters introduces significant memory constraints. Traditional PCA algorithms require loading all data into memory simultaneously, which becomes infeasible when analyzing thousands of deep neural network models with hundreds of millions to billions of parameters. In such settings, directly constructing the covariance matrix is computationally expensive and memory-prohibitive. To address this, we exploit the dual PCA formulation, in which principal directions are recovered from the eigen-decomposition of the Gram matrix rather

than the covariance of the features (Schölkopf et al., 1998; Shawe-Taylor et al., 2005). This approach has been extended to functional and multivariate settings, where the dual eigenproblem provides a scalable approximation to the spectra of covariance operators (Golovkine et al., 2026). By projecting the data into the space spanned by the $n_{\text{models}}$ samples instead of the original $n_{\text{params}}$ features, the dimensionality is reduced from $n_{\text{params}} \times n_{\text{params}}$ to $n_{\text{models}} \times n_{\text{models}}$; mathematically, this is equivalent to standard PCA because the nonzero eigenvalues of the covariance matrix $XX^\top$ and the Gram matrix $X^\top X$ coincide, and the principal components in the original space can be reconstructed from the sample-space eigenvectors. To further scale PCA to extremely high-dimensional models, we combine this dual formulation with randomized numerical linear algebra. Specifically, the eigendecomposition of the Gram matrix is computed using a randomized SVD scheme, which reduces computational cost while preserving spectral accuracy (Halko et al., 2011). Since storing full datasets or full parameter vectors is infeasible, both covariance and Gram matrices are constructed incrementally. We build on the principles of incremental and streaming PCA algorithms (Ross et al., 2008; Cardot et al., 2018), adapting them to extremely high-dimensional model parameters with micro-batch accumulation and GPU-accelerated matrix operations. Model parameters are streamed from disk in batches, enabling PCA on datasets that exceed available memory. Our method performs PCA in four stages (D.1): (1) incremental estimation of the empirical mean, (2) streamed construction of the Gram matrix, (3) randomized eigendecomposition, and (4) vectorized recovery of the principal components in the original parameter space. This results in a scalable PCA framework suitable for analyzing collections of models with billions of parameters, even when the complete dataset cannot fit in memory. Table 9 demonstrates that dual PCA can be effectively used to accurately generate weight spaces for models such as ResNet18 and ViT-Small-192, with parameter counts on the order of $O(10M)$, as well as for larger models like BERT-Base with up to $O(100M)$ parameters.

Table 9: *Flow Matching Hyperparameters and Performance Results For 100 generated samples projected to 98-99 PCA components using dual PCA*

| Model | Hidden Dim | Time Embed | Org. Scores | Avg Score |
|---|---|---|---|---|
| *ResNet18 (dataset: CIFAR-10, metric: accuracy %)* | | | | |
| ResNet18 | 1024 | 128 | 94.45 ± 0.14 | 93.52 ± 0.16 |
| *ViT-Small-192 (dataset: CIFAR-10, metric: accuracy %)* | | | | |
| ViT-Small-192 | 512 | 64 | 83.30 ± 0.29 | 83.83 ± 0.1 |
| *BERT-Base (dataset: Yelp, metric: Spearman's correlation)* | | | | |
| BERT-Base | 1024 | 64 | 0.7902 ± 0.0061 | 0.7909 ± 0.005 |

### D.2 NOTATION AND ALGORITHM

Let $W = [w_1, \ldots, w_n] \in \mathbb{R}^{d \times n}$ denote the weight matrix where $n$ is the number of trained models, $d$ is the number of parameters per model, $k$ is the number of principal components to retain, and $w_i \in \mathbb{R}^d$ is the $i$-th model's flattened weights. Let $\tilde{W} = W - \mu \mathbf{1}^\top \in \mathbb{R}^{d \times n}$ denote the centered weight matrix where $\mu = \frac{1}{n} \sum_{i=1}^n w_i$ is the empirical mean.

The algorithm consists of four sequential passes:

1. Incremental Mean Computation: Compute the empirical mean in batches to avoid loading all models into memory:
$$\mu = \frac{1}{n} \sum_{i=1}^n w_i$$

2. Gram Matrix Construction: Build the $n \times n$ Gram matrix block-wise, exploiting GPU parallelism while keeping only two micro-batches in GPU memory at a time:
$$G_{ij} = (w_i - \mu)^\top (w_j - \mu), \quad i, j = 1, \ldots, n$$

3. Randomized Eigendecomposition: Compute the top $k$ eigenvectors of $G$ using randomized SVD (Halko et al., 2011):
$$G \approx U\Sigma U^\top, \quad U \in \mathbb{R}^{n \times k}, \quad \Sigma = \text{diag}(\sigma_1, \ldots, \sigma_k)$$

where $\sigma_i$ are singular values. Since $G = \tilde{W}^\top \tilde{W}$ is symmetric, eigenvalues are $\lambda_i = \sigma_i^2$.

4. Principal Components in Parameter Space: Recover components in the original $d$-dimensional space via back-projection:

$$P = \tilde{W}U \in \mathbb{R}^{d \times k}$$

Components are computed using GPU-accelerated matrix multiplication and normalized to unit length.

### D.2.1 COMPLEXITY ANALYSIS

Time complexity per pass:

- Incremental Mean Computation: $\mathcal{O}(nd)$ — single pass through all data
- Gram Matrix Construction: $\mathcal{O}(n^2 d)$ — compute $n^2$ pairwise inner products
- Randomized SVD: $\mathcal{O}(n^2 k)$ — randomized SVD with 5 iterations
- Principal Components in Parameter Space: $\mathcal{O}(ndk)$ — back-project to $k$ components

Complexity is practically limited by $\mathcal{O}(n^2 d)$ when $k < n \ll d$, dominated by Gram matrix construction.

### D.2.2 EMPIRICAL TIMING ANALYSIS

We conducted a comprehensive timing study of our pipeline using a single NVIDIA A100 40GB GPU to understand the computational costs of each phase. We analyzed the end-to-end timing for three representative architectures - ResNet18 (11M parameters), ViT-Small-192 (5.5M parameters), and BERT-Base (118M parameters), each trained on 100 models. All experiments were run on a single NVIDIA A100 GPU with FP16 precision for Dual PCA implementation. The timing estimates can be found in Table 10 and Table 11.

Table 10: *Setup Phase Timing Breakdown on NVIDIA A100*

| Model | Canonicalization | PCA Fitting | Flow Training | Total Setup |
|---|---|---|---|---|
| ResNet18 (11M) | 144s | 60s | 66s | **270s** |
| ViT-Small (2.84M) | 1,002s | 12s | 60s | **1,074s** |
| BERT-Base (118M) | 6,900s | 360s | 66s | **7,322s** |

Setup phase is executed once per model collection (100 models) and prepares the system for subsequent model generation.

Table 11: *Generation Phase Timing per Single Model on NVIDIA A100*

| Model | Latent Flow | Inverse PCA | Inference[a] | Total |
|---|---|---|---|---|
| ResNet18 (11M) | 0.032s | 0.049s | 1.68s[b] | **1.76s** |
| ViT-Small (2.84M) | 0.031s | 0.015s | 1.633s | **1.67s** |
| BERT-Base (118M) | 0.150s | 1.60s | 20s | **21.75s** |

[a] Inference includes WSO reconstruction, model loading, and evaluation on test set.
[b] ResNet18 inference time includes BatchNorm recalibration

### D.2.3 SCALABILITY DISCUSSION

The dual PCA formulation is particularly advantageous when $d \gg n$, as the Gram matrix $G \in \mathbb{R}^{n \times n}$ is much smaller than the $d \times d$ covariance matrix required by standard PCA. This reduces both computational cost (from $\mathcal{O}(nd^2)$ to $\mathcal{O}(n^2 d)$ for covariance construction) and memory requirements (from $\mathcal{O}(d^2)$ to $\mathcal{O}(n^2)$). With modern high-memory GPUs (e.g., NVIDIA H100 with 80GB HBM3) and FP16 precision, the micro-batch size $m$ can be tuned to balance GPU memory constraints and computational efficiency. The FP16 option effectively doubles these capacity limits while introducing negligible numerical error. As GPU memory and compute continue to improve, we expect this approach to scale naturally to even larger model collections.

# E   DATASET GENERATION

Table 12 and Table 13 provide the details of the architecture and training hyperparameters used to create the trained neural network datasets that were used to train DeepWeightFlow. The training datasets can be made available on request.

Table 12: *Hyperparameters for training the neural networks that were used as the training datasets for DeepWeightFlow. Final weights for each seed after the epochs listed in the table are treated as a single datapoint. We train 100 such models, using early stopping to halt training when validation performance plateaus.*

| Model | Dataset | Params | LR Schedule | Optimizer | LR | Weight Decay | Batch Size | Epochs |
|---|---|---|---|---|---|---|---|---|
| MLP | Iris | 131 | None | Adam | 1e-3 | 0 | 16 | 100 |
| MLP | MNIST | 26.5K | None | Adam | 1e-3 | 0 | 64 | 5 |
| MLP | Fashion | 118K | None | AdamW | 1e-3 | 0 | 128 | 25 |
| SmallCNN | CIFAR-10 | 12.4K | None | AdamW | 1e-3 | 1e-3 | 128 | 50 |
| ResNet-18 | STL-10 | 11.2M | Warmup+Cosine | SGD | 0.1 | 5e-4 | 128 | 10 |
| ResNet-18 | CIFAR-10 | 11.2M | Cosine | SGD | 0.1 | 5e-4 | 128 | 100 |
| ResNet-20 | CIFAR-10 | 0.27M | None | Adam | 1e-3 | 0 | 128 | 5 |
| Vit-Small-192 | CIFAR-10 | 2.8M | Cosine | AdamW | 3e-4 | 0.05 | 128 | 300 |
| BERT-Base | Yelp Review | 118M | None | AdamW | 1e-4 | 0 | 32 | 3 |

Table 13: *Model architectures for the neural networks used to train DeepWeightFlow. For the MLPs, the first number in the Architecture definition is the input dimension. For the ResNets, "blocks" refer to residual blocks. For training BERT models, we use only a subset of the YelpReview dataset for training and testing for this experiment.*

| Model | Architecture | Parameters | Dataset | Input Dim |
|---|---|---|---|---|
| MLP | [4, 16, 3] | 131 | Iris | $4 \times 150$ |
| MLP | [784, 32, 32, 10] | 26,506 | MNIST | $28 \times 28$ |
| MLP | [784, 128, 128, 10] | 117,770 | Fashion-MNIST | $28 \times 28$ |
| SmallCNN | 3 conv, 2 FC | 12,042 | CIFAR-10 | $32 \times 32 \times 3$ |
| ResNet-20 | $3 \times [3, 3, 3]$ blocks | 272,474 | CIFAR-10 | $32 \times 32 \times 3$ |
| ResNet-18 | $4 \times [2, 2, 2, 2]$ blocks | 11.17M | CIFAR-10 | $32 \times 32 \times 3$ |
| ResNet-18 | $4 \times [2, 2, 2, 2]$ blocks | 11.17M | STL-10 | $96 \times 96 \times 3$ |
| Vit-Small-192 | 194 embedding dimension, 6 blocks, 3 heads | 2.87M | CIFAR-10 | $32 \times 32 \times 3$ |
| BERT-Base | 768 embed dim, 12 blocks, 12 heads | 118M | Yelp Review | 128 tokens |

The ResNet-20 neural networks used have notably lower parameter counts than the ResNet-18 neural networks, as the former is narrower while being deeper to reduce model complexity in training for smaller datasets. The ResNet-18 configuration is typical (He et al., 2016). The specific block layouts are described in Table 13.

# F   HYPERPARAMETERS OF DEEPWEIGHTFLOW MODELS

In Table 14 we provide the hyperparameters of the DeepWeightFlow models. The FM model architecture varies by the dimensionality of the neural network weights in the training set and their architecture.

# G   COMPUTATIONAL EFFICIENCY: TRAINING AND GENERATION TIME

DeepWeightFlow demonstrates significant computational advantages over existing parameter generation methods. We compare our approach with RPG (Wang et al., 2025), the current state-of-the-art in recurrent parameter generation, across multiple architectures and configurations.

When incorporating Git Re-basin (Ainsworth et al., 2023) for weight alignment, the additional computational overhead is minimal:

- ResNet-18: 2 minutes for aligning 100 models
- Vit-Small-192 (Transfusion): 13 minutes for aligning 100 models

The results in Table 15 show that DeepWeightFlow consistently generates high-quality models while having lower training and inference time on similar GPUs.

Table 14: *DeepWeightFlow Flow Matching training hyperparameters*

| Parameter | Value | Parameter | Value |
|---|---|---|---|
| **Architecture** | | **Training** | |
| Flow Model Hidden Dims | $[d_h, d_h/2, d_h]^a$ | Optimizer | AdamW |
| Time Embedding Dim | 4–128[b] | Learning Rate | $5 \times 10^{-4}$ / $1 \times 10^{-4}$[h] |
| Activation Function | GELU | Weight Decay | $1 \times 10^{-5}$ |
| Layer Normalization | Yes | AdamW $\beta$ | (0.9, 0.95) |
| Dropout Rate | 0.1–0.4[c] | Batch Size | 2–8[d] |
| **Flow Matching** | | **Training** | |
| Time Distribution | Uniform / Beta[i] | Training Iterations | 30,000 |
| Noise Scale ($\sigma$) | 0.001 | Training Data Size | 100 models |
| Source Distribution | $\mathcal{N}(0, \sigma_s^2 I)^e$ | LR Scheduler | CosineAnnealing |
| | | $\eta_{\min}$ | $1 \times 10^{-6}$ |
| **Generation** | | **Preprocessing** | |
| ODE Solver | Runge-Kutta 4 | Weight Matching | Git Re-Basin/TransFusion[f] |
| Integration Steps | 100 | PCA Method | Incremental/Dual PCA[j] |
| Generated Samples | 25–100[k] | BN Recalibration | ResNets only[g] |

[a] $d_h \in \{32, 64, 128, 256, 384, 512, 1024\}$ depending on architecture complexity
[b] Time embedding: 4 for Iris MLP, 64 for ResNet-20/MNIST/Fashion-MNIST/Vit-Small-192/BERT-Base, 128 for ResNet-18
[c] Dropout: 0.4 for Iris MLP, 0.1 for all other architectures
[d] Batch size: 2 for BERT-Base, 4 for Vit-Small-192, 8 for all others
[e] $\sigma_s = 0.001$ for Vit-Small-192 and BERT-Base, $\sigma_s = 0.01$ for all other architectures
[f] Git Re-Basin for ResNets/MLPs, TransFusion for Vision Transformers and BERT
[g] BatchNorm statistics recalibrated using test data only for ResNet architectures post-generation
[h] Learning rate: $1 \times 10^{-4}$ for BERT-Base, $5 \times 10^{-4}$ for all others
[i] Time distribution: Beta(2,5) for BERT-Base, Uniform for all others
[j] PCA: Incremental PCA (scikit-learn) for ResNet-18/Vit-Small-192; GPU-accelerated Dual PCA (Gram matrix, FP16) for BERT-Base
[k] Generated samples: 25 for Vit-Small-192, 100 for all other architectures

## H CHOOSING THE RIGHT SOURCE DISTRIBUTION

The choice of source distribution for these generative models has a significant impact on the performance of the generated models. Table Table 16 highlights the importance of selecting a source distribution that aligns well with the target distributions to ensure reliable and high-quality weight generation.

## I DIVERSITY OF THE GENERATED NEURAL NETWORKS

In Table 17, we provide the numerical estimates of mIoU, the Jensen-Shannon, Wasserstein, and Nearest Neighbors (NN) distances between generated and original neural networks, highlighting the diversity of the generated neural networks.

## J FINETUNING MODELS FOR TRANSFER LEARNING ON UNSEEN DATASETS

We leverage ResNet-18 models trained and generated on the CIFAR-10 dataset to adapt to other unseen datasets, specifically STL-10 and SVHN (Table 6). We first evaluate the performance of the generated CIFAR-10 models on these datasets without any fine-tuning (Epoch 0). Subsequently, we fine-tune the models using the standard training set of the target dataset and evaluate them on the corresponding test set. Fine-tuning is performed for up to 5 epochs using the AdamW optimizer with a learning rate of $1 \times 10^{-4}$, weight decay of $1 \times 10^{-4}$, and a cosine learning rate scheduler with $T_{\max} = epochs$ for smooth decay. We use a detach ratio of 0.4 (same as used by Saragih et al. (2025b)) and the cross-entropy loss is used as the objective function. We further experiment with SmallCNN models generated for the STL-10 dataset and transfer them to the CIFAR-10 dataset in a similar fashion, comparing our results with those reported by Schürholt et al. (2024).

Table 15: *Performance comparison between DeepWeightFlow, RPG, P-diff, and D2NWG (Wang et al., 2025; 2024; Soro et al., 2025). RPG generates a single neural network per run, while DeepWeightFlow generates neural networks sequentially in a single workflow. D2NWG and P-diff only generate 2048 weights within the pretrained ResNet18 backbone (Soro et al., 2025).*

| Model | Method | Hidden Dim | Training Time | Generation Time (1 model) | GPU |
|---|---|---|---|---|---|
| ResNet-18 (11.7M params) | RPG (sequential)[†] | - | - | 18.6 min | H100 |
| | RPG (partially parallel)[†] | - | - | 1.8 min | H100 |
| | RPG (fully parallel)[†] | - | - | 1.7 min | H100 |
| | DeepWeightFlow[§] | 1024 | 3 min | **1.38 seconds** | A100 |
| | DeepWeightFlow + rebasin[§] | 1024 | 2 min + 3 min | **1.38 seconds** | A100 |
| | P-diff[¶] | - | - | 3 hours[*] | - |
| | D2NWG[¶] | - | - | 1.5 hours[*] | - |
| ViT-Tiny (5M params) | RPG (flatten)[‡] | - | 6.2 hours | 9.8 min | H100 |
| | RPG (by channel)[‡] | - | 14.2 hours | 9.8 min | H100 |
| | RPG (within layer)[‡] | - | 6.2 hours | 9.8 min | H100 |
| | RPG (partially parallel)[†] | - | - | 1.1 min | H100 |
| | RPG (fully parallel)[†] | - | - | 1.1 min | H100 |
| Vit-Small-192 (2.8M params) | DeepWeightFlow[§] | 256 | 21 min | 2.16 seconds | A100 |
| | DeepWeightFlow[§] | 384 | 19 min | **1.70 seconds** | H100 |
| | DeepWeightFlow + transfusion[§] | 384 | 13 min + 19 min | **1.70 seconds** | H100 |

[†] Available RPG inference times from Wang et al. (2025).

[‡] RPG training + sequential inference time from Wang et al. (2025) (Table 4 and Table 18); numbers available for single neural network generation.

[§] DeepWeightFlow performs sequential generation of models. Numbers reported here are for ResNet-18 generated using standard incremental PCA and ViT-Small-192 for training and generation without PCA.

[¶] P-diff and D2NWG perform only partial generation of 2048 weights within a pretrained backbone (Soro et al., 2025) (Table 11).

[*] P-diff and D2NWG times reported are likely for generating 100 models; divide by 100 for approximate per-model time (P-diff: 1.8 min/model, D2NWG: 0.9 min/model).

Table 16: *Evaluating the impact of various source distribution choices in FM mapping on the performance of complete weights generated by DeepWeightFlow.*

| Model & Source Distribution | With Rebasin (%) | Without Rebasin (%) |
|---|---|---|
| **Vit-Small-192 on CIFAR-10** | | |
| Original Accuracy | | $83.29 \pm 0.29$ |
| Gaussian(0, 0.01) | $78.31 \pm 10.99$ | $76.69 \pm 14.37$ |
| Gaussian(0, 0.001) | **$82.90 \pm 0.70$** | $82.40 \pm 5.29$ |
| **MLP on MNIST** | | |
| Original Accuracy | | $96.32 \pm 0.20$ |
| Kaiming Initialization | $81.33 \pm 14.10$ | $67.35 \pm 26.10$ |
| Gaussian(0, 0.01) | **$96.18 \pm 0.23$** | **$96.22 \pm 0.22$** |

ViT: Architecture: Vit-Small-192 (2.7M parameters), Dataset: CIFAR-10, Flow Hidden Dim: 384, Time Embed Dim: 64

MLP: Architecture: MLP (26.5K parameters), Dataset: MNIST, Flow Hidden Dim: 256, Time Embed Dim: 64 Dropout: 0.1

In these experiments, we evaluate three approaches: (1) random initialization (baseline), (2) direct transfer from original pretrained models from the source dataset, and (3) transfer from flow-generated models trained on the source weight distribution. All models are fine-tuned on the target dataset using the same protocol described above. Our results demonstrate that flow-generated models achieve comparable or occasionally slightly superior performance to the original pretrained models when transferred to the target domain. This validates that our flow matching approach successfully captures the essential characteristics of the learned weight distributions, producing high-quality models that preserve transferable features from the source task. The competitive performance of generated models relative to their pretrained counterparts confirms that the flow-based generative process maintains the representational quality necessary for effective transfer learning.

Table 17: *Comparison of 100 complete neural network weights generated by DeepWeightFlow with and without Git Re-Basin through maximum Intersection over Union (IoU), Jensen-Shannon, Wasserstein, and Nearest Neighbors (NN) distances. For MNIST, we use MLP with $d_h = 512$ and $10\%$ dropout. For CIFAR-10, we use ResNet-18 with $d_h = 1024$. Lower scores indicate closer relationships. (Org. - original, Gen. - generated)*

| Dataset/Architecture | Metric | Org. to Org. | Org. to Gen. | Gen. to Org. | Gen. to Gen. |
|---|---|---|---|---|---|
| *MNIST - MLP* | | | | | |
| DeepWeightFlow w/ Re-Basin | IoU | - | - | $0.8187 \pm 0.0385$ | - |
| | Wasserstein | - | 13.4125 | 21.2867 | 11.6721 |
| | Jensen-Shannon | - | 0.7146 | 0.8326 | 0.7146 |
| | NN | $23.0393 \pm 0.2214$ | $9.7232 \pm 10.4398$ | $1.7526 \pm 0.1671$ | $11.7407 \pm 10.5471$ |
| | Cosine Sim. | 0.1962 | 0.2093 | 0.2093 | 0.2157 |
| | $L^2$ | 25.5268 | 25.2278 | 25.2278 | 25.1367 |
| DeepWeightFlow w/o Re-Basin | IoU | - | - | $0.8256 \pm 0.0748$ | - |
| | Wasserstein | - | 15.1185 | 25.6979 | 17.6939 |
| | Jensen-Shannon | - | 0.8181 | 0.8326 | 0.7293 |
| | NN | $27.4895 \pm 0.2007$ | $12.3710 \pm 12.4410$ | $1.7916 \pm 0.3753$ | $9.7956 \pm 11.2484$ |
| | Cosine Sim. | 0.0088 | 0.0187 | 0.0187 | 0.0189 |
| | $L^2$ | 28.3513 | 28.1681 | 28.1681 | 28.2423 |
| *CIFAR-10 - ResNet-18* | | | | | |
| DeepWeightFlow w/ Re-Basin | IoU | - | - | $0.6289 \pm 0.0160$ | - |
| | Wasserstein | - | 15.1236 | 27.5994 | 20.3590 |
| | Jensen-Shannon | - | 0.8242 | 0.8326 | 0.8242 |
| | NN | $27.9643 \pm 0.0841$ | $13.3136 \pm 14.0490$ | $0.3649 \pm 0.0836$ | $7.9625 \pm 12.6314$ |
| | Cosine Sim. | 0.2497 | 0.2542 | 0.2542 | 0.2570 |
| | $L^2$ | 28.9520 | 28.8494 | 28.8494 | 28.8105 |
| DeepWeightFlow w/o Re-Basin | IoU | - | - | $0.6314 \pm 0.0198$ | - |
| | Wasserstein | - | 16.7654 | 29.8754 | 20.9545 |
| | Jensen-Shannon | - | 0.5018 | 0.8326 | 0.7014 |
| | NN | $30.2421 \pm 0.0766$ | $13.4767 \pm 14.8165$ | $0.3667 \pm 0.0590$ | $9.3245 \pm 13.7908$ |
| | Cosine Sim. | 0.1754 | 0.1818 | 0.1818 | 0.1832 |
| | $L^2$ | 30.3523 | 30.2332 | 30.2332 | 30.2922 |

### J.1 TRANSFER LEARNING FOR DATASETS WITH DIFFERENT NUMBERS OF CLASSES

We evaluate the transferability of flow-generated neural network weights by leveraging ResNet-18 models trained and generated on the CIFAR-10 dataset to adapt to the CIFAR-100 dataset in Table 18, which presents a significantly more challenging task with 100 classes compared to CIFAR-10's 10 classes. We compare three approaches: (1) random initialization (baseline), (2) direct transfer from original CIFAR-10 pretrained models, and (3) transfer from flow-generated models as described above. For all pretrained approaches, we replace the final fully-connected layer to accommodate the 100-class output and reinitialize it using Kaiming initialization. We first assess zero-shot performance (Epoch 0), where models are evaluated on CIFAR-100 without any fine-tuning beyond the FC layer adaptation. Subsequently, we fine-tune the models for 1, 5, and 10 epochs using few-shot learning with 50 samples per class from CIFAR-100 dataset. Fine-tuning is performed using the AdamW optimizer with a learning rate of $1 \times 10^{-3}$, weight decay of $1 \times 10^{-4}$, and a cosine annealing learning rate scheduler with $T_{\max}$ set to the number of epochs. The cross-entropy loss is used as the objective function. This experimental setup allows us to assess whether flow-generated models preserve transferable representations learned from CIFAR-10 and can effectively adapt to the more challenging CIFAR-100 classification task, demonstrating the quality and utility of our generative weight modeling approach.

## K CONDITIONAL GENERATION WITH MODIFIED DEEPWEIGHTFLOW

### K.1 MULTI-CLASS GENERATION WITH DEEPWEIGHTFLOW

To demonstrate the ability of DeepWeightFlow to generalize across tasks, we show conditional generation across datasets by operating directly in weight space with simple time and class embeddings at the flow model input (Lipman et al., 2023). The models displayed in Table 19 are different from the MLPs described in Appendix E in that they have equal weight space sizes and an identical architecture.

### K.2 MULTI-CLASS AND MULTI-ARCHITECTURE CONDITIONAL GENERATION

To adapt DeepWeightFlow for multi-class and multi-architecture conditional generation, we incorporated a class embedding MLP to produce dense class embeddings, which are concatenated with

Table 18: *Zero-shot performance at epoch 0 and fine-tuning results for* **complete** *ResNet-18 parameters trained on CIFAR-10 and transferred to the CIFAR-100 dataset. The parameters come from DeepWeight-Flow, SANE (Schürholt et al., 2024), RandomInit, and a Pretrained Transfer baseline. RandomInit denotes a fresh Kaiming-He initialization. Pretrained denotes models first trained on CIFAR-10 and then transferred to CIFAR-100. Generated denotes parameters sampled from the respective generative model. Models pretrained on CIFAR-10 (10 classes) have their classification head replaced to accommodate CIFAR-100's 100 classes during transfer learning, while retaining the learned convolutional features. Best scores for each fine-tuning setting are shown in bold.*

| Epoch | Model | Method | CIFAR-100 |
|---|---|---|---|
| 0 | SANE | tr. fr. scratch | $1.00 \pm 0.00$ |
| | | Finetuned | $1.0 \pm 0.3$ |
| | | $SANE_{SUB}$ | $\mathbf{1.1 \pm 0.2}$ |
| | DeepWeightFlow | RandomInit | $0.98 \pm 0.06$ |
| | | Pretrained | $1.01 \pm 0.17$ |
| | | Generated | $1.06 \pm 0.26$ |
| 1 | SANE | tr. fr. scratch | $17.5 \pm 0.7$ |
| | | Finetuned | $25.7 \pm 1.3$ |
| | | $SANE_{SUB}$ | $26.9 \pm 1.4$ |
| | DeepWeightFlow | RandomInit | $23.36 \pm 1.05$ |
| | | Pretrained | $37.03 \pm 1.34$ |
| | | Generated | $\mathbf{38.37 \pm 1.15}$ |
| 5 | SANE | tr. fr. scratch | $36.5 \pm 2.0$ |
| | | Finetuned | $45.7 \pm 1.0$ |
| | | $SANE_{SUB}$ | $45.6 \pm 1.2$ |
| | DeepWeightFlow | RandomInit | $56.79 \pm 0.69$ |
| | | Pretrained | $\mathbf{67.39 \pm 0.38}$ |
| | | Generated | $67.37 \pm 0.53$ |

Table 19: Multiclass DeepWeightFlow generation results without PCA compression and with Git Re-Basin.

| Dataset | Original | Generated |
|---|---|---|
| MNIST | $96.74 \pm 0.25$ | $96.61 \pm 0.22$ |
| Fashion-MNIST | $86.80 \pm 0.31$ | $86.46 \pm 0.28$ |

the input and time embeddings. These combined vectors are then fed into the flow model. We began by training a single flow matching model to generate weights for MNIST and Fashion-MNIST datasets using an MLP architecture that is identical across both datasets. By conditioning on these class embeddings, the single flow model successfully generated weights that achieved good performance for both datasets.

Next, we attempted to train DeepWeightFlow to learn multiple classes in the full-rank weight space, which requires that the models have identical parameter counts. While full-rank learning across multiple classes proved difficult, using PCA-reduced weight space allowed the model to handle multiple classes and architectures simultaneously. However, the generated models did not achieve extremely high accuracy, as seen in Table 20. A key reason is that FM models perform best when the weight space distribution is smooth and consistent. Introducing multiple architectures or datasets fragments this space, making it challenging for a single learned flow to interpolate or extrapolate correctly. This remains a work in progress.

Table 20: *Conditional Multiclass Cross-Architecture Generation with PCA Compression. Shows 4 classes across distinct architectures. DeepWeightFlow trained with all classes canonicalized. All values are mean ± standard deviation. Models were generated with PCA compression.*

| Class (Dataset) | Original | Generated |
|---|---|---|
| Class 0 (MNIST) | $96.78 \pm 0.23$ | $54.11 \pm 23.88$ |
| Class 1 (Fashion-MNIST) | $86.82 \pm 0.33$ | $43.21 \pm 19.65$ |
| Class 2 (Iris) | $70.23 \pm 9.29$ | $53.03 \pm 17.37$ |
| Class 3 (ResNet20-CIFAR10) | $73.62 \pm 2.24$ | $50.90 \pm 31.24$ |

