# OpenReview forum: "DeepWeightFlow: Re-Basined Flow Matching for Generating Neural Network Weights"
_ICLR.cc/2026/Conference — ICLR 2026 Poster_

### Official Review · Reviewer_xfHk · 2025-10-28

**Soundness:** 2
**Presentation:** 2
**Contribution:** 3
**Rating:** 4
**Confidence:** 4

**Summary:**

The submission presents a flow-based method for neural network weight generation. The authors identify generation quality, speed and diversity as key metrics to improve. Their proposed method extends existing work with a flow-based generation method, that demonstrates high quality of generated weights, on par with the original weights without the need for fine-tuning at considerably faster training and generation times. The authors also successfully make use of model canonicalization and PCA to reduce the complexity and size of the domain.

**Strengths:**

- The topic of the submission is a highly relevant topic, fast and high fidelity generation remains unsolved. The application of flows on weights is well motivated as a match for high fidelity in high dimensions at relatively high speed.
- The experimental evaluation show that the proposed method does indeed generate weights that perform very well, to the level of the original models. At the same time, training the flow models appears to be significantly faster than previous work. While these numbers are hidden in the appendix, they present a significant improvement in training speed (and potentially generation speed) over previous work. The experiment setup here is not entirely clear to me, but if this holds up in practice it seems a big deal for weight generation. For that reason I'm somewhat baffled the authors chose not to highlight it.
- I particularly appreciated the evaluation on canonicalization, in which regimes it has an effect and where it seems to diminish.
- The authors make an effort to review the related work and contextualize their method well.

**Weaknesses:**

While I overall like the proposed method, and it seems to address limitations of previous work, mostly in quality and speed of generation, there are weaknesses that prevent me from scoring the submission higher at this point. I encourage the authors to address them, and am willing to improve my score.

- Quality, model size and generation speed are relevant problems, and need to be considered in new weight generation methods. The experiments focus on evaluating original vs generated model performance as a proxy for quality, where previous works like D2NWG or RPG [2, 3] appear to perform similarly. The authors further argue with differences in the experimental setup or generation speed, but I could only find generation times for RPG and DeepFlowNets in the Appendix. This comparison appears to take RPG generation times and extrapolate by 100. I can't find any reference numbers to match the author's, so I would ask them to make their calculations transparent. I understand there is an argument to be made for parallelization, but I in general don't see the value of generating 100 models for the same task. If parallelization is the strength here, then I'd ask the authors to add the 1 model vs 100 model generation to extract that signal cleanly.
- As the authors note, experiment details matter for parameter synthesis and have made comparisons in previous work challenging. Therefore I would ask them to make their experimental setup clearer. On what and how many model checkpoints is DeepWeightFlow trained? What varies between the checkpoints in the dataset per experiment? Is there one global flow model, or one model per experiment?
- Where and how is PCA used exactly? If using PCA is the main driver for training and generation speeds, I'd encourage the authors to run ablations so that readers understand where the speed improvements come from. If - as claimed - PCA provides a general compression of significant quality, then improvements should carry over to other methods. That said, previous work using auto encoders generally only achieved high compression ratios for small datasets or datasets with relatively low amount of diversity (shared seeds, etc), so I'm somewhat skeptical that linear compression can achieve as much.
- The authors use the IoU to show diversity. However, it seems from their experiments that the generated models are substantially less diverse than the original models. While it's in the same diversity range as previous parameter generation methods [1], the experiments aren't directly comparable. Notably, the diversity trends from original to generated are reversed. This does not need to be an issue, but if the authors aim at diversity, this requires some discussion. Also, the authors chose to show diversity on small MLPs and MNIST. Since scalability is one of the main arguments, I encourage the authors to evaluate diversity on larger models and harder tasks.
- Ultimately, the goal for weight generation has to be to go beyond the original models. This can mean generalization beyond the original data, but might also be a recombination of dataset/architecture/size/knowledge. Here, the contribution is rather narrow. Other work on weight generation has trained on diverse model datasets to cover a broader distribution or introduced generation conditioned on data [3, 4, 5]. I would encourage the authors to discuss how their model helps make progress towards real-world applications.
- Minor issues in the related work: Permutation symmetries have been used as data augmentations for general in [6, 7]. Graph-based models are used successfully in the INR community, but [2,8] aren't graph auto encoders as far as I can tell. Neither seem to use any connectivity information.
- NIT: Commonly, hypernetworks mean generator-only models that are trained to synthesize the weights of a target model using target model signal. That seems different from the representation learning methods that operate on weights and only use weight signals.

[1] Wang et al, 2024: Neural Network Diffusion.
[2] Wang et al., 2025: Scaling Up Parameter Generation: A Recurrent Diffusion Approach.
[3] Soro et al, 2024: Diffusion-Based Neural Network Weights Generation.
[4] Falk et al., 2025: The Impact of Model Zoo Size and Composition on Weight Space Learning.
[5] Falk et al, 2025: Learning Model Representations Using Publicly Available Model Hubs.
[6] Schuerholt et al, 2021: Self-Supervised Representation Learning on Neural Network Weights.
[7] Peebles et al, 2022: Learning to Learn with Generative Models of Neural Network Checkpoints.
[8] Schuerholt et al, 2022: Hyper-Representations as Generative Models: Sampling Unseen Neural Network Weights.

**Questions:**

See weaknesses above.

---

> ### Author Response · Authors · 2025-11-28
>
> We would like to thank the reviewer for their time and effort in reviewing the draft. We have tried to address the concerns as follows:
>
> W1: We understand the concern and have instead shown the generation time for a single neural network for all models in Table 15. The numbers for the other models have been taken directly from the relevant papers. We have added numbers for P-diff and D2NWG. However, it is to be noted that the latter two report numbers for generating 100 neural networks, and we have left the numbers as such with a footnote. Also, they perform only partial weight generation and not complete weight generation, as RPG and our model.
>
> W2: We thank the reviewer, as this helps us add clarity to the draft. We have added text in section 4, page 5, in the description of the training dataset to specify that we did not use checkpoints from a single training run. In fact, all neural networks in the training set were endpoints of training runs performed with completely distinct randomly seeded initializations. This helps with the heterogeneity of the training set and allows us to avoid mode collapse. DeepWeightFlow is not a global model as it is not conditioned. We train individual DeepFlowWeight models for each architecture and each dataset. We have added a note to the first paragraph of section 5 in page 6 to this end. Given the very low training time and the ability to transfer learn as shown in the draft, we chose to take this path. However, we have also added results on conditional training in Appendix K and left a note in the conclusion proposing this as future work.
>
> W3: We have used PCA to preprocess the training data and inverse PCA during the generation of the neural networks. We have added a paragraph in section 4  to clarify this and added a lot more details in Appendix D. We have added a table in Appendix D, Table 8, to show that training time is significantly reduced by using incremental PCA. We have also added results throughout the draft (notably table 5) for generating the BERT 118M parameter model using Dual PCA.
>
> W4: We thank the reviewer for pointing this out. We have added results for the diversity of generated neural networks for ResNet-18 trained on CIFAR-10. The results can be found in Table 17
>
> W5: We explore conditional generation and add it to Appendix K, showing that our model works well in this paradigm too. We have tried multi-dataset multi-architecture too, and have added it to Appendix K with the caveat that it's our initial exploration. The results are not too promising, and hence we leave this for future work
>
> W6 & W7: We thank the reviewer for pointing this out. We have changed the relevant sentence in the related works section
>
> We have made several modifications as a result of the comments left by the reviewer. We are confident this has improved the quality of our work and hope that the reviewer agrees to the same.

---

### Official Review · Reviewer_mm9F · 2025-10-29

**Soundness:** 3
**Presentation:** 4
**Contribution:** 2
**Rating:** 6
**Confidence:** 5

**Summary:**

This paper introduces DeepWeightFlow, a flow-matching approach that generates complete neural-network weights. Using optional Incremental PCA for tractability, it scales from small models to architectures with roughly 10M parameters, delivering performance that matches or exceeds prior methods on the same benchmarks.

**Strengths:**

**Originality**

* Direct Flow Matching in weight space (faster than diffusion).
* Symmetry-aware via Git Re-Basin / TransFusion.
* Lean scaling with (Incremental) PCA no learned autoencoder.

**Quality**

* Solid results across MLP/ResNet/ViT and datasets.
* Clear ablations (capacity vs. canonicalization; source distribution).
* Practical touches (BN stats recalibration) that boost reliability.

**Clarity**

* Well-structured, accessible explanations

# Significance

* Large efficiency gains: Cuts weight generation from hours to minutes, making high-quality ensemble creation computationally practical.
* Better starting points: Provides strong initializations that accelerate fine-tuning and enhance transfer learning.
* Scalable foundation:  A robust, scalable framework for generative modeling in weight space that can serve as a strong baseline for future,extensions.

**Weaknesses:**

* **Unconditional generation limits controllability:**
  The method’s unconditional design restricts its ability to generalize or transfer across datasets and architectures. While presented as “data-free,” this property also constrains the model’s flexibility

* **Inability to jointly encode across datasets:**
  Because the generator is unconditional, it cannot represent or sample from mixtures of weights trained on diverse datasets. Each dataset requires its own independently trained flow.

* **Overstated scalability claims:**
  The paper refers to networks of roughly *10M parameters* as “large-scale.” While this is nontrivial, it falls short of the scales (hundreds of millions to billions of parameters) typically considered large in contemporary neural network research, leaving true scalability uncertain.

* **Unclear handling of label or output-size mismatches:**
  When transferring to datasets with a different number of output classes, it is not specified how the method manages mismatched output dimensions or classifier heads. This omission raises questions about generalization to tasks with differing label structures.

**Questions:**

For more question see weaknesses

1.  **Transfer Learning Baseline:** Why does Table 5 not include the transfer performance of the original models from your training set as a baseline? This would clarify if DeepWeightFlow generates models with inherently better transferability.

2.  **Scaling to Billion-Parameter Models:** How do you foresee the IPCA-based approach scaling to billion-parameter models? Is a single linear projection via PCA sufficient and computationally tractable for such a high-dimensional and complex weight space?

3.  **Rationale for Unconditional Generation:** What was the motivation for focusing on unconditional generation? What are the main challenges you see in extending DeepWeightFlow to a conditional framework for sampling models with specific attributes?

4.  **Generalization Beyond Classification:** Could you comment on the expected challenges of applying DeepWeightFlow to tasks beyond classification (e.g., in NLP or regression)? Would the current canonicalization methods be sufficient for the different symmetries in those domains?

---

> ### Author Response · Authors · 2025-11-28
>
> We thank the reviewer for the work they put into writing the review of our paper. We find the questions and concerns quite pertinent and try to address them both in the forum and in the updated draft.
>
> W1: We agree with the reviewer. In our work, we tried to address unconditional generation using flow matching models as a novel concept, as conditional generation has been addressed using diffusion models (such as RPG, D2NWG) and, to a certain extent, in flow matching models by the authors of FLoWN. Our model does equally well or better than these models. However, given that both model training and neural network generation are a lot efficient with our model, we alleviate the problem of having to train multiple models to a certain extent. That being said, we have now explored conditioning and present some results in Appendix K.
>
> W2: We added some results in Appendix K and left a note in the conclusion, noting that we leave this to future work.
>
> W3: We thank the reviewer for pointing this out. We have changed the text in several places to tone down our enthusiasm. Moreover, we have added results for the generation BERT 118M parameter model and show both high efficiency and good quality of generation. The results can be found in Table 5. We have also added a discussion about scaling with Dual PCA in Appendix D. Given that we use only one A100 40GB GPU and the constraint of our method is mostly from memory consumption, we are confident we can scale to larger models with larger GPUs or multi-GPU training. We leave this for future work.
>
> W4: We thank the reviewer for pointing this out. We indeed did not try datasets that have a different number of classes. This can be achieved by swapping out the classification layers with the correct dimensional ones, but would require significant fine-tuning. Given the ease with which we can train the flow matching model (very low training time), we chose to take the path of having individual DeepFlowWeight models for individual datasets, with some possibility of transfer learning when the number of classes is the same. That being said, we have added new results to Table 6 to compare with SANE. In Appendix J, we have added clarity to how we perform transfer learning and added new results in Table 18 on transfer learning on the CIFAR-100 dataset using a CIFAR-10 trained ResNet-18 backbone and comparing it with SANE. The neural networks generated by our model perform better than those generated by SANE or randomly initialized models.
>
> Q1: We appreciate the comment and have added the results to Table 6 and Table 18
>
> Q2: We have used Dual PCA for scaling to the BERT 118M model and have added that to the draft. Please see section 4, page 5, and Appendix D. We can use Dual PCA for larger models and have discussed scaling in Appendix D.
>
> Q3: Please see our comment above in W1
>
> Q4: We appreciate the comment. We have added results for the BERT 118M parameter model on a regression task (Yelp review prediction). The results can be seen in Table 5.
>
> The comments have helped us improve our work, and we have added several new results. We hope the reviewer appreciates them.

---

### Official Review · Reviewer_ZyLw · 2025-11-01

**Soundness:** 3
**Presentation:** 3
**Contribution:** 3
**Rating:** 6
**Confidence:** 3

**Summary:**

This work proposes an approach to generating complete sets of neural network parameters using an MLP generator trained via a flow matching objective. Training data was obtained purely by training many separate models on datasets like MNIST, CIFAR10, SVHN, etc. The proposed method is trained to generate weights for different architectures including MLPs, ResNets, and ViTs.

Specifically, this method uses incremental PCA to reduce the effective dimensionality of the target weight set, then performs flow matching in that space, followed by an inverse transformation (of the pca reduction) to go back to the original weight space. Permutation symmetry is addressed via cannonicalization using appraoches such as git re-basin and transfusion.

The approach is evaluated on the above datasets for in-distribution performance, as well as oof performance on CIFAR10 -> SVHN and STL-10. The diversity of generated samples is evaluated in terms of their predictive behavior.

**Strengths:**

* I appreciate that the approach is a sensible way of handling the high dimensional weight space i.e. iterative pca -> flow matching.

* I found the paper well-written and easy to follow

* I am impressed that this work is able to handle complete weight sets even with more complicated architectures like ViTs, and those that include batchnorm.

* The results of this work are compelling compared to other previous work like p-diff and flown.

**Weaknesses:**

* I'm confused about the effect of cannonicalization. Ostensibly it makes training easier but the results shown in figure 2 don't seem to reveal any difference in final test accuracy of generated samples. What is the effect of not using the (computationally expensive) process)?

* I think diversity could be addressed in more than just behavioral statistics. We might also want to know about the average L2 distance between generated and training networks. We might also want to see if for any (or the best) generated network, is there an exact or near-exact copy of it in the training corpus. These datasets are often quite small and memorization is common here.

**Questions:**

1) How does this method scale in terms of absolute VRAM numbers? This is usually the limiting factor with these weight generation methods. At some point we need to do a matrix multiplication where one dimension is the full parameter vector. But with this incremental PCA approach scaling might be better?

---

> ### Author Response · Authors · 2025-11-28
>
> We thank the reviewer for the insights they provide. We have addressed the concerns raised as follows:
>
> W1: Thanks for pointing this out. The outcome of canonicalization is shown in Table 4. We find that flow matching models with lower complexity benefit from canonicalization for the generation of smaller neural networks. However, there are some advantages to using canonicalization for larger neural networks, and we confirm this with an additional experiment with the BERT 118M parameter model. We have added that result to Table 5. We have also discussed the impact of canonicalization in section 5.1, page 8.
>
> W2: We appreciate the comment and have added L2 and cosine similarity to Table 17 in Appendix I. Moreover, we have added results for ResNet-18 trained on CIFAR-10 data to show diversity in a larger model with a different architecture.
>
> Q1: We have used a single GPU (A100-40GB) for all neural network generation and training of the FM models. We used incremental PCA for models of O(10M) parameters and Dual PCA for models of O(100M) parameters to overcome the limitations of memory (please see section 4, page 5, and appendix D). We can scale further by using GPUs with larger VRAM. We have explored the possibility of multi-GPU implementation, but leave the details for future work. Hence, we are confident that our FM model can scale to larger neural networks.
>
> The reviewer's comments have led to significant additions and modifications to the draft that we feel have improved the clarity and quality of the work. We hope the reviewer feels as positive about these as us.

---

### Official Review · Reviewer_8Vjc · 2025-11-01

**Soundness:** 3
**Presentation:** 4
**Contribution:** 3
**Rating:** 6
**Confidence:** 5

**Summary:**

This submission presents a novel approach for weight generation capable of assembling full neural network models from a diverse set of neural network model checkpoints. This is accomplished using a flow matching approach, which is coined "DeepWeightFlow" by the authors. This approach works directly in the weight space and employs model alignment techniques such as Git Re-basin or TransFusion.

Experimental results are reported on models trained on tabular and computer vision datasets. In-distribution evaluations are done against SOTA approaches, where each SOTA approach is trained on ResNet-18 models trained on CIFAR-10, aiming to sample complete ResNet-18 models for CIFAR-10. The same is done for ViT models trained on CIFAR-10 dataset. Out-of-distribution evaluations are done on models trained on multiple datasets, but only for the presented approach and not compared against SOTA approaches except FLoWN. Multiple ablation studies are reported to understand what impact different components of the presented approach have on sampling performance.

I very much like the presented work and think that the in-distribution results are quite impressive, since it evaluates the generated model's performance without fine-tuning. This is very impressive given the diverse model collection used for training of the presented approach. This is particularly important since some SOTA approaches like RGP do generate weights that are similar to the very homogeneous model collection they have been trained on, and break when randomly initialised models are used for training. I truly appreciate the author's sensitivity to this detail in their experimental setup (line 200-202, line 248-251).

Regardless of this, for this submission, I have some concerns given the experimental setup, which are (i) the limitation to only one computer vision dataset in the comparison against other SOTA approaches (Table 2) and missing broad evaluation of other SOTA approaches in their out-of-distribution evaluation (Table 5). I will outline all details in the strengths, weaknesses, and questions section below.

Overall, I think this work provides a sufficient level of novelty and contribution to be accepted, but I would like to wait to have my questions and concerns addressed, i.e., I would be willing to increase my rating during the discussion with the authors.

**Strengths:**

- **(S1)**: The proposed method is able to generate neural networks that do not require any further fine-tuning. I think this is impressive, given the reported performance being close to the original performance of the model dataset. I am speaking specifically about the results in Table 2 and Table 3. I would have some questions regarding generalizability, though. These are outlined in the questions section.

- **(S2)**: I very much appreciate the emphasis of this work on diversity in the model dataset being used for training. I think this is important for the entire model weight generation community. Speaking about this, I think that the results in Table 2 are a little bit misleading as the numbers reported by Wang et al 2025 (=95.1) have been generated by finetuned trajectories of the same seeded neural network model (this is already mentioned multiple times by the authors in the submission). Having said this, I think that the results of this work (=93.47) are more impressive since they are generated using a much more difficult setup, i.e., a more diverse model dataset for training, as the results from Wang et all 2025 (=95.1). I am not sure about the results of Soro et al (2025), to the best of my knowledg,e they only partially generate ResNet weights and do not fully generate a ResNet model. Finally, the results of Schuerholt et al (2024) also have some leakage of information as they use samples from the target model dataset to fit their KDE for sampling.

- **(S3)**: I also appreciate additional results reported on the diversity of generated models and sampling efficiency. This provides some additional information contextualizing the proposed method.

- **(S4)**: It is great to see a reproducibility statement in this work. The weight generation community needs more of this.

- **(S5)**: This paper is a nice read since it is easy to read and follow.

**Weaknesses:**

- **(W1)**: As mentioned above, in Table 2, I would like to see more comparisons beyond CIFAR-10 results. I think that might provide a more general impression of the proposed methods' performance.

- **(W2)**: In the same sense, for Table 5, I would like to see comparisons against other SOTA methods. This would help to understand the proposed method's capabilities in the context of related work.

- **(W3)**: Since the authors additionally claim that the proposed method is able to scale, I would love to see some scalability ablation results going beyond ViT-Small-192.

**Questions:**

I have some questions:

- **(Q1)**: This is a short one. Why does the paper mention that the ResNet-18 transfer learning experiments are done using PCA as preprocessing (Section 5.2), and for the in-domain experiments, this is not mentioned (Section 5.1)?

- **(Q2)**: The proposed approach works directly in weight space or in PCA-reduced weight space (as far as I understood it). The paper also mentioned that sampling can be done "without the requirement of additional conditioning during training or inference" (line 308-309). How do you make sure the method does not overfit to the given training data? Or said the other way around: how homogeneous does your model dataset need to be to perform well?

- **(Q3)**: In a similar spirit, I see that your model dataset is trained only with one specific set of hyperparameters, which is different to, e.g., the model zoos [1] used in some other work. Don't understand me wrong, I appreciate the focus of this work on diversity (wrt. seeds) of your model dataset, but I would be interested in your thoughts on choosing it this way. [1] Model Zoos: A Dataset of Diverse Populations of Neural Network Models, NeurIPS, 2022

- **(Q4)**: What exactly is used for training the underlying FM backbone? The model dataset for training is outlined nicely in the appendix, but how many checkpoints and which checkpoints are you using for the training of your approach (maybe I missed it)

- **(Q5)**: Table 9 in the appendix shows the number of parameters of the model datasets. Why is the ResNet-20 (0.27M parameter) smaller than the ResNet-18 ()11.2M parameter? I guess this is a typo, right?

- **(Q6)**: I would be wondering if the proposed method is able to generalize beyond one image dataset, i.e, what would happen if the proposed method were trained on model datasets trained on CIFAR-10 and FashionMNIST and MNIST. Would this help in transfer learning scenarios?

- **(Q7)**: Same question wrt.. the architecture. Is the proposed method able to generalize to multiple architectures? I mean not to train one FM model per neural network architecture, but to train the proposed methods with multiple architectures. Similarly, is the proposed method able to generate neural networks of different architectures with the same learned FM model?

- **(Q8)**: I think that the result reported for Soro et al (2025) should be moved to partially generate ResNet model weights and not fully generated ResNet model weights. Please correct me if I am wrong.

- **(Q9)**: Finally, you provide efficiency results for RPG but not for D2NWG. Both provide code for their method. Is there a reason why you do not provide numbers for D2NWG?

As I already said, I think this is fascinating, and I would be happy to increase my rating once I fully understand the generalizability capabilities of this approach.

**Details Of Ethics Concerns:**

-

---

> ### Author Response · Authors · 2025-11-28
>
> We thank the reviewer for the time and effort they put into evaluating our work. We have found the questions and concerns to be quite relevant and have made several modifications to our work and added additional results. We have also modified the text to reflect the same and added details where the text lacked clarity. The following are the point-by-point discussions of the review:
>
> W1: We have added results for STL-10 with ResNet-18 and compared them with the results reported by P-diff, FLoWN, and NM. It is to be noted that the latter three generate partial weights, while we generate complete weights with the same accuracy as the training set. This can be found in the new Table 3 on page 7.
>
> W2: For our work on transfer learning, we tried to focus on a comparison of ResNet models, and FLoWN is the only one we came across. Other papers have discussed transfer learning (such as Schürholt et al. 2022), but on a very simple CNN from the model zoo. We have now added a transfer learning comparison with SANE using the same CNN that the authors used. This can be found in Table 6 on page 9. In addition, we have added a comparison on transfer learning using our original dataset that our model was trained on, and show equivalent performance. We have added this to Table 6.
>
> W3: We have added results for BERT (118M parameters) in Table 5 in page 7 and have added discussions on how we use Dual PCA (page 5 and appendix D) to achieve this and how this is scalable further. BERT hyperparameters have been added to Appendix E. We would like to note that scaling is possible using multiple GPUs for models of O(1B) parameters or more, but we have not demonstrated it. We leave this for future work. We have added some notes in Appendix D on scalability.
>
> Q1: PCA was used for both in-domain and transfer learning. We have added a short note on page 5, Section 4, for scaling using PCA.
>
> Q2: We would like to note that the training dataset is not derived from the checkpoint of a single training round but from distinct random initializations leading to significant diversity in the training set. We have added this to the part on training data generation in section 4 on page 5. We do not observe mode collapse in the generated networks, as also seen in section 5.2
>
> Q3: We tried to show our results on various models, such as simple MLPs, ResNets, ViTs, and have added BERT. Our primary reason for not using Model Zoo was our wish to have control over the generation of datasets, including the ability to vary the initialization schemes of the trained dataset. This also gave us a comprehensive estimate of the resources and time required for the full training and inference cycle.
>
> Q4: We did not use checkpoints. We used distinct randomly initialized models in the training data. We have added text to clarify this in the main paper at the end of section 4 on page 5.
>
> Q5: ResNet-20 is a deeper but narrower model, often adapted from the ResNet-18 architecture for smaller image sizes. Hence, the number of parameters is much less. We have added this note to Appendix E, page 22.
>
> Q6: This is a good question, and we have not tried this in the current work, as our focus was training and generation in individual datasets and transfer learning. We would love to try this out in future work. However, we have added some results on conditional generation in Appendix K and have added a short paragraph in the conclusions, leaving it to future work.
>
> Q7: We have added some results in Appendix K with the note that the results are preliminary, and we leave it to future work to improve the results.
> .
> Q8: Thank you for the suggestion. We have done so. Please note the modification in Table 2
>
> Q9: Thanks for pointing this out. We have added results for P-diff and D2NWG and show that our model is more efficient than both. The results can be found in Table 15 in Appendix G.
>
> The comments and concerns of the reviewer have led us to add new results to the paper and provide clarification of several points. We feel this has increased the quality of our work significantly and hope the reviewer agrees with our additions and modifications.

---

### Author Response · Authors · 2025-11-28

We thank all the reviewers for their comments and concerns. We have addressed them in the new version of the draft and also added some new results that were motivated by the comments of the reviewers. We have responded to the reviewers individually and welcome further comments and discussions.

---

> ### Author Response · Authors · 2025-12-03
> **Overview of the modifications and additions we have made in response to the reviewer feedback**
>
> We addressed all the issues raised by the reviewers in the first round of reviews. We added several experiments and showed that DeepWeightFlow can scale to even larger neural networks than we had shown in the first draft. We added tables and subsections to describe all the new experiments and to elaborate on the points that the reviewers pointed out lacked clarity. We expanded the appendix we technical details of our experiments and have provided code to reproduce our results.
>
>
> **Major concerns raised by reviewers and our response**
>
> Below is a list of the major concerns that the reviewers raised and our response to them is indented in
>
> - Comparisons with other SOTA models were a bit narrow, limited mostly to CIFAR10 and were missing broader benchmarks as well as limited comparisons for transfer tasks.
>     - We added new experiments with other datasets including STL10. We added comparisons with P-diff, FLoWN, NM, D2NWG.
>
>
> - Scaling claims required evidence beyond small and mid scale models and reviewers asked for clarity on memory limits and PCA based scaling.
>
>     -  We added BERT experiments to demonstrate scaling up to 100M parameters (10x scaling) and provided reasoning of why we believe we can scale even further using Dual PCA.
>
> - Training data construction and checkpoint usage needed more clarity including how many checkpoints were used, why model zoo was not used, and why PCA was applied differently in different sections.
>
>     - We clarified the training data construction, especially that checkpoints are not reused and that datasets come from distinct random initializations.
>     -  We added full explanations on scaling using incremental PCA and Dual PCA along with concrete numbers and discussion of multi GPU potential. We expanded both the main text and the appendix.
>
> - Canonicalization impact was unclear and reviewers wanted quantitative evidence of its benefit.
>
>     - We added explanations on the effects of canonicalization and responded to the reviewer explaining the effects of canonicalization
>
> - Missing baselines for transfer learning and missing efficiency numbers for methods like D2NWG.
>
>     - We added experiments to compare transfer learning against more benchmarks like SANE and transfer learning when the number of dataset target classes are different.
>
> - Questions about architecture heterogeneity, dataset heterogeneity, and conditional generation.
>
>     - Expanded diversity evaluation by adding L2 and cosine similarity metrics and added new experiments results on larger models such as ResNet18.
>
> - The paper only explores unconditional flow matching
>
>     - We added notes on architectural and dataset generalization including preliminary cross architecture experiments using conditional generation. Added more numbers around efficiency as asked including D2NWG numbers. Additions were made to the appendix and conclusion.
>
> We hope the above modifications and additions sufficiently address the concerns raised by the reviewers. We would like to note that two of the reviewers expressed willingness to raise the score on satisfactory review response. We are very grateful to the reviewers and we strongly believe that the quality of our work has improved because of the reviews, the additional results that we added, and the clarifications that we provided in the draft and in review response. We thank the ACs for taking up this complex task under unusual circumstances.

---

### Meta-Review · Area_Chair_mrSk · 2026-01-07

**Summary:**

The rebuttal addressed the concerns raised by the reviewers and provided comprehensive experiments and analysis of the proposed method. It helps assess the paper's contributions. Three reviewers recommend acceptance with a marginally above the acceptance threshold after discussion, and the AC concur. The final version should include all reviewer comments, suggestions, and additional experiments from the rebuttal.

**Reviewer Concerns:**

The authors' rebuttal addressed the reviewers' questions.

**Reviewer Scores:**

The reviewers' scores reflect the contributions of this work.

---

### Decision · Program_Chairs · 2026-01-26

Accept (Poster)